# Homeostatic reinforcement learning for integrating reward collection and physiological stability

Mehdi Keramati[1,2]*, Boris Gutkin[1,3]*

[1]Group for Neural Theory, INSERM U960, Département des Etudes Cognitives, Ecole Normale Supérieure, PSL Research University, Paris, France; [2]Gatsby Computational Neuroscience Unit, University College London, London, United Kingdom; [3]Center for Cognition and Decision Making, National Research University Higher School of Economics, Moscow, Russia

**Abstract** Efficient regulation of internal homeostasis and defending it against perturbations requires adaptive behavioral strategies. However, the computational principles mediating the interaction between homeostatic and associative learning processes remain undefined. Here we use a definition of primary rewards, as outcomes fulfilling physiological needs, to build a normative theory showing how learning motivated behaviors may be modulated by internal states. Within this framework, we mathematically prove that seeking rewards is equivalent to the fundamental objective of physiological stability, defining the notion of physiological rationality of behavior. We further suggest a formal basis for temporal discounting of rewards by showing that discounting motivates animals to follow the shortest path in the space of physiological variables toward the desired setpoint. We also explain how animals learn to act predictively to preclude prospective homeostatic challenges, and several other behavioral patterns. Finally, we suggest a computational role for interaction between hypothalamus and the brain reward system.

*For correspondence: mehdi@gatsby.ucl.ac.uk (MK); boris.gutkin@ens.fr (BG)

**Competing interests:** The authors declare that no competing interests exist.

**Reviewing editor**: Eve Marder, Brandeis University, United States

## Introduction

Survival requires living organisms to maintain their physiological integrity within the environment. In other words, they must preserve homeostasis (e.g. body temperature, glucose level, etc.). Yet, how might an animal learn to structure its behavioral strategies to obtain the outcomes necessary to fulfill and even preclude homeostatic challenges? Such, efficient behavioral decisions surely should depend on two brain circuits working in concert: the hypothalamic homeostatic regulation (HR) system, and the cortico-basal ganglia reinforcement learning (RL) mechanism. However, the computational mechanisms underlying this obvious coupling remain poorly understood.

The previously developed classical negative feedback models of HR have tried to explain the hypothalamic function in behavioral sensitivity to the 'internal' state, by axiomatizing that animals minimize the deviation of some key physiological variables from their hypothetical setpoints (*Marieb & Hoehn, 2012*). To this end, a direct corrective response is triggered when a deviation from setpoint is sensed or anticipated (*Sibly & McFarland, 1974*; *Sterling, 2012*). A key lacuna in these models is how a simple corrective action (e.g. 'go eat') in response to a homeostatic deficit might be translated into a complex behavioral strategy for interacting with the dynamic and uncertain external world.

On the other hand, the computational theory of RL has proposed a viable computational account for the role of the cortico-basal ganglia system in behavioral adaptation to the 'external' environment, by exploiting experienced environmental contingencies and reward history (*Sutton & Barto, 1998*; *Rangel et al., 2008*). Critically, this theory is built upon one major axiom, namely, that the objective of

**eLife digest** Our survival depends on our ability to maintain internal states, such as body temperature and blood sugar levels, within narrowly defined ranges, despite being subject to constantly changing external forces. This process, which is known as homeostasis, requires humans and other animals to carry out specific behaviors—such as seeking out warmth or food—to compensate for changes in their environment. Animals must also learn to prevent the potential impact of changes that can be anticipated.

A network that includes different regions of the brain allows animals to perform the behaviors that are needed to maintain homeostasis. However, this network is distinct from the network that supports the learning of new behaviors in general. These two systems must, therefore, interact so that animals can learn novel strategies to support their physiological stability, but it is not clear how animals do this.

Keramati and Gutkin have now devised a mathematical model that explains the nature of this interaction, and that can account for many behaviors seen among animals, even those that might otherwise appear irrational. There are two assumptions at the heart of the model. First, it is assumed that animals are capable of guessing the impact of the outcome of their behaviors on their internal state. Second, it is assumed that animals find a behavior rewarding if they believe that the predicted impact of its outcome will reduce the difference between a particular internal state and its ideal value. For example, a form of behavior for a human might be going to the kitchen, and an outcome might be eating chocolate.

Based on these two assumptions, the model shows that animals stabilize their internal state around its ideal value by simply learning to perform behaviors that lead to rewarding outcomes (such as going into the kitchen and eating chocolate). Their theory also explains the physiological importance of a type of behavior known as 'delay discounting'. Animals displaying this form of behavior regard a positive outcome as less rewarding the longer they have to wait for it. The model proves mathematically that delay discounting is a logical way to optimize homeostasis.

In addition to making a number of predictions that could be tested in experiments, Keramati and Gutkin argue that their model can account for the failure of homeostasis to limit food consumption whenever foods loaded with salt, sugar or fat are freely available.

behavior is to maximize reward acquisition. Yet, this suite of theoretical models does not resolve how the brain constructs the reward itself, and how the variability of the internal state impacts overt behavior.

Accumulating neurobiological evidence indicates intricate intercommunication between the hypothalamus and the reward-learning circuitry (*Palmiter, 2007*; *Yeo & Heisler, 2012*; *Rangel, 2013*). The integration of the two systems is also behaviorally manifest in the classical behavioral pattern of anticipatory responding in which, animals learn to act predictively to preclude prospective homeostatic challenges. Moreover, the 'good regulator' theoretical principle implies that 'every good regulator of a system must be a model of that system' (*Conant & Ashby, 1970*), accentuating the necessity of learning a model (either explicit or implicit) of the environment in order to regulate internal variables, and thus, the necessity of associative learning processes being involved in homeostatic regulation.

Given the apparent coupling of homeostatic and learning processes, here, we propose a formal hypothesis for the computations, at an algorithmic level, that may be performed in this biological integration of the two systems. More precisely, inspired by previous descriptive hypotheses on the interaction between motivation and learning (*Hull, 1943*; *Spence, 1956*; *Mowrer, 1960*), we suggest a principled model for how the rewarding value of outcomes is computed as a function of the animal's internal state, and of the approximated need-reduction ability of the outcome. The computed reward is then made available to RL systems that learn over a state-space including both internal and external states, resulting in approximate reinforcement of instrumental associations that reduce or prevent homeostatic imbalance.

The paper is structured as follows: After giving a heuristic sketch of the theory, we show several analytical, behavioral, and neurobiological results. On the basis of the proposed computational integration of the two systems, we prove analytically that reward-seeking and physiological stability are two sides of the same coin, and also provide a normative explanation for temporal discounting of

reward. Behaviorally, the theory gives a plausible unified account for anticipatory responding and the rise-fall pattern of the response rate. We show that the interaction between the two systems is critical in these behavioral phenomena and thus, neither classical RL nor classical HR theories can account for them. Neurobiologically, we show that our model can shed light on recent findings on the interaction between the hypothalamus and the reward-learning circuitry, namely, the modulation of dopaminergic activity by hypothalamic signals. Furthermore, we show how orosensory information can be integrated with internal signals in a principled way, resulting in accounting for experimental results on consummatory behaviors, as well as the pathological condition of over-eating induced by hyperpalatability. Finally, we discuss limitations of the theory, compare it with other theoretical accounts of motivation and internal state regulation, and outline testable predictions and future directions.

## Results

### Theory sketch

A self-organizing system (i.e. an organism) can be defined as a system that opposes the second law of thermodynamics (*Friston, 2010*). In other words, biological systems actively resist the natural tendency to disorder by regulating their physiological state to fall within narrow bounds. This general process, known as homeostasis (*Cannon, 1929*; *Bernard, 1957*), includes adaptive behavioral strategies for counteracting and preventing self-entropy in the face of constantly changing environments. In this sense, one would expect organisms to reinforce responses that mitigate deviation of the internal state from desired 'setpoints'. This is reminiscent of the drive-reduction theory (*Hull, 1943*; *Spence, 1956*; *Mowrer, 1960*) according to which, one of the major mechanisms underlying reward is the usefulness of the corresponding outcome in fulfilling the homeostatic needs of the organism (*Cabanac, 1971*). Inspired by these considerations (i.e. preservation of self-order and reduction of deviations), we propose a formal definition of primary reward (equivalently: reinforcer, economic utility) as the approximated ability of an outcome to restore the internal equilibrium of the physiological state. We then demonstrate that our formal homeostatic reinforcement learning framework accounts for some phenomena that classical drive-reduction was unable to explain.

We first define 'homeostatic space' as a multidimensional metric space in which each dimension represents one physiologically-regulated variable (the horizontal plane in *Figure 1*). The physiological state of the animal at each time $t$ can be represented as a point in this space, denoted by $H_t = (h_{1,t}, h_{2,t}, .., h_{N,t})$, where $h_{i,t}$ indicates the state of the $i$-th physiological variable. For example, $h_{i,t}$ can refer to the animal's glucose level, body temperature, plasma osmolality, etc. The homeostatic setpoint, as the ideal internal state, can be denoted by $H^* = (h_1^*, h_2^*, .., h_N^*)$. As a mapping from the physiological to the motivational state, we define the 'drive' as the distance of the internal state from the setpoint (the three-dimensional surface in *Figure 1*):

$$D(H_t) = \sqrt[m]{\sum_{i=1}^{N} \left| h_i^* - h_{i,t} \right|^n} \tag{1}$$

$m$ and $n$ are free parameters that induce important nonlinear effects on the mapping between homeostatic deviations and their motivational consequences. Note that for the simple case of $m = n = 1$, the drive function reduces to Euclidian distance. We will later consider more general nonlinear mappings in terms of classical utility theory. We will also discuss that the drive function can be viewed as equivalent to the information-theoretic notion of *surprise*, defined as the negative log-probability of finding an organism in a certain state ($D(H_t) = -\ln p(H_t)$).

Having defined drive, we can now provide a formal definition for primary reward. Let's assume that as the result of an action, the animal receives an outcome $o_t$ at time $t$. The impact of this outcome on different dimensions of the animal's internal state can be denoted by $K_t = (k_{1,t}, k_{2,t}, .., k_{N,t})$. For example, $k_{i,t}$ can be the quantity of glucose received as a result of outcome $o_t$. Hence, the outcome results in a transition of the physiological state from $H_t$ to $H_{t+1} = H_t + K_t$ (See *Figure 1*) and thus, a transition of the drive state from $D(H_t)$ to $D(H_{t+1}) = D(H_t + K_t)$. Accordingly, the rewarding value of this outcome can be defined as the consequent reduction of drive:

$$r(H_t, K_t) = D(H_t) - D(H_{t+1})$$
$$= D(H_t) - D(H_t + K_t) \tag{2}$$

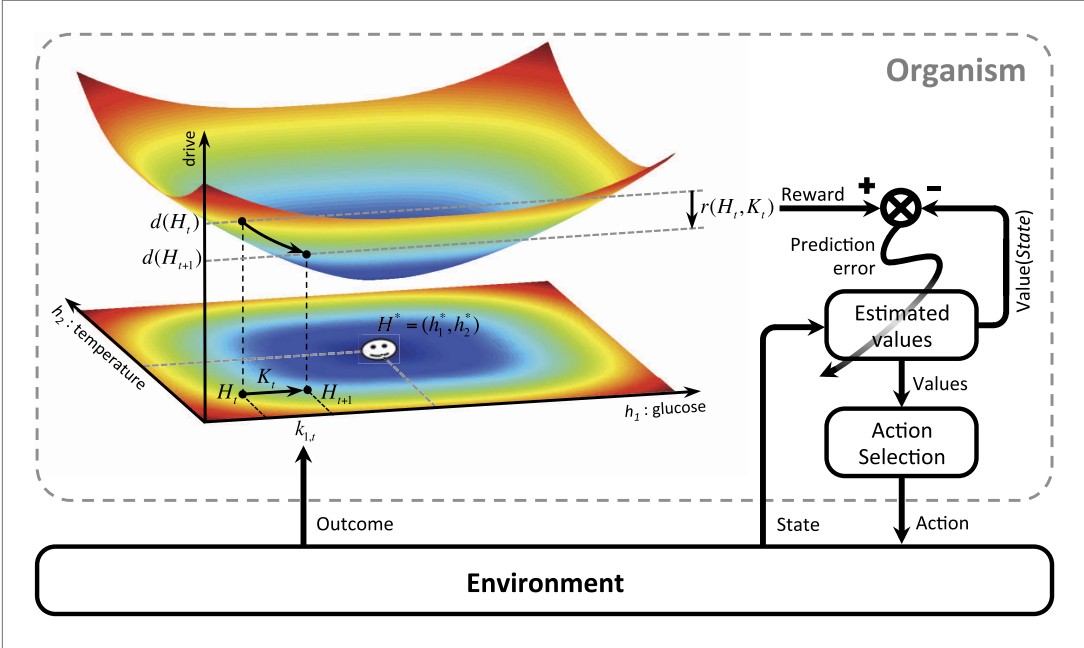

**Figure 1**. Schematics of the model in an exemplary two-dimensional homeostatic space. Upon performing an action, the animal receives an outcome $K_t$ from the environment. The rewarding value of this outcome depends on its ability to make the internal state, $H_t$, closer to the homeostatic setpoint, $H^*$, and thus reduce the drive level (the vertical axis). This experienced reward, denoted by $r(H_t, K_t)$, is then learned by an RL algorithm. Here a model-free RL algorithm is shown in which a reward prediction error signal is computed by comparing the realized reward and the expected rewarding value of the performed response. This signal is then used to update the subjective value attributed to the corresponding response. Subjective values of alternative choices bias the action selection process.

Intuitively, the rewarding value of an outcome depends on the ability of its constituting elements to reduce the homeostatic distance from the setpoint or equivalently, to counteract self-entropy. As discussed later, the additive effect ($K_t$) of these constituting elements on the internal state can be approximated by the orosensory properties of outcomes. We will also discuss how erroneous estimation of drive reduction can potentially be a cause for maladaptive consumptive behaviors.

We hypothesize in this paper that the primary reward constructed as proposed in *Equation 2* is used by the brain's reward learning machinery to structure behavior. Incorporating this physiological reward definition in a normative RL theory allows us to derive one major result of our theory, which is that the rationality of behavioral patterns is geared toward maintaining physiological stability.

## Rationality of the theory

Here we show that our definition of reward reconciles the RL and HR theories in terms of their normative assumptions: reward acquisition and physiological stability are mathematically equivalent behavioral objectives. More precisely, given the proposed definition of reward and given that animals discount future rewards (*Chung & Herrnstein, 1967*), any behavioral policy, $\pi$, that maximizes the sum of discounted rewards (*SDR*) also minimizes the sum of discounted deviations from the setpoint, and vice versa. In fact, starting from an initial internal state $H_0$, the sum of discounted deviations (*SDD*) for a certain behavioral policy $\pi$ that causes the internal state to move in the homeostatic space along the trajectory $p(\pi)$, can be defined as:

$$SDD_\pi\left(H_0\right) = \int_{p(\pi)} \gamma^t . D\left(H_t\right) . dt \tag{3}$$

Similarly, the sum of discounted rewards (SDR) for a policy $\pi$ can be defined as:

$$SDR_\pi\left(H_0\right) = \int_{p(\pi)} \gamma^t . r_t . dt = \int_{p(\pi)} \gamma^t . \left(D\left(H_t\right) - D\left(H_{t+dt}\right)\right) . dt \qquad (4)$$

It is then rather straightforward to show that for any initial state $H_0$, we will have (See 'Materials and methods' for the proof):

$$if \quad \gamma < 1: \quad \operatorname*{argmin}_\pi SDD_\pi\left(H_0\right) = \operatorname*{argmax}_\pi SDR_\pi\left(H_0\right) \qquad (5)$$

where $\gamma$ is the discount factor. In other words, the same behavioral policy satisfies optimal reward-seeking as well as optimal homeostatic maintenance. In this respect, reward acquisition sought by the RL system is an efficient means to guide an animal's behavior toward fulfilling the basic objective of defending homeostasis. Thus, our theory suggests a physiological basis for the rationality of reward seeking.

## Normative role of temporal discounting

In the domain of animal behavior, one fundamental question is why animals should discount rewards the further they are in the future. Our theory indicates that reward seeking without discounting (i.e., if $\gamma = 1$) would not lead, and may even be detrimental, to physiological stability (See 'Materials and methods'). Intuitively, this is because a future-discounting agent would always tend to expedite bigger rewards and postpone punishments. Such an agent, therefore, tries to reduce homeostatic deviations (which is rewarding) as soon as possible, and thus, tries to find the shortest path toward the setpoint. A non-discounting agent, in contrast, can always compensate for a deviation-induced punishment by reducing that deviation any time in the future.

While the formal proof of the necessity of discounting is given in the 'Materials and methods', let us give an intuitive explanation. Imagine you had to plan a 1-hr hill walk from a drop-point toward a pickup point, during which you wanted to minimize the height (equivalent to drive) summed over the path you take. In this summation, if you give higher weights to your height in the near future as compared to later times, the optimum path would be to descend the hill and spend as long as possible at the bottom (i.e. homeostatic setpoint) before returning to the pickup point. *Equation 5* shows that this optimization is equivalent to optimizing the total discounted rewards along the path, given that descending and ascending steps are defined as being rewarding and punishing, respectively (*Equation 2*).

In contrast, if at all points in time you give equal weights to your height, then the summed height over path only depends on the drop and pickup points, since every ascend can be compensated with a descend at any time. In other words, in the absence of discounting, the rewarding value of a behavioral policy that changes the internal state only depends on the initial and final internal states, regardless of its trajectory in the homeostatic space. Thus, when $\gamma = 1$, the values of any two behavioral policies with equal net shifts of the internal state are equal, even if one policy moves the internal state along the shortest path, whereas the other policy results in large deviations of the internal state from the setpoint and threatens survival. These results hold for any form of temporal discounting (e.g., exponential, hyperbolic). In this respect, our theory provides a normative explanation for the necessity of temporal discounting of reward: to maintain internal stability, it is necessary to discount future rewards.

## A normative account of anticipatory responding

A paradigmatic example of behaviors governed by the internal state is the anticipatory responses geared to preclude perturbations in regulated variables even before any physiological depletion (negative feedback) is detectable. Anticipatory eating and drinking that occur before any discernible homeostatic deviation (*Woods & Seeley, 2002*), anticipatory shivering in response to a cue that predicts the cold (*Mansfield et al., 1983*; *Hjeresen et al., 1986*), and insulin secretion prior to meal initiation (*Woods, 1991*), are only a few examples of anticipatory responding.

One clear example of a conditioned homeostatic response is animals' progressive tolerance to ethanol-induced hypothermia. Experiments show that when ethanol injections are preceded (i.e., are predictable) by a distinctive cue, the ethanol-induced drop of the body core temperature of animals diminishes along the trials (*Mansfield & Cunningham, 1980*). *Figure 2* shows that when the temperature was measured 30, 60, 90, and 120 min after daily injections, the drop of temperature below the baseline was significant on the first day, but gradually disappeared over 8 days. Interestingly, in the

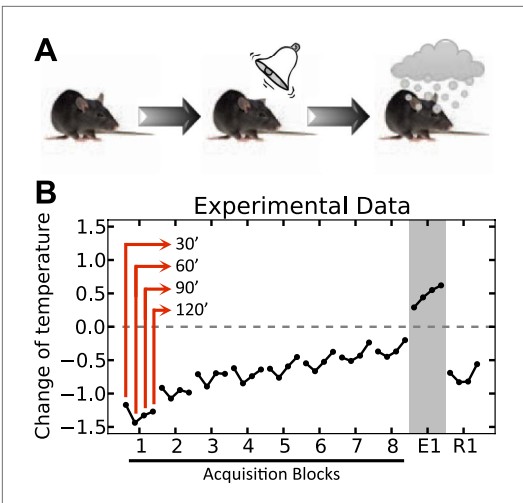

**Figure 2**. Experimental results (adapted from *Mansfield & Cunningham, 1980*) on the acquisition and extinction of conditioned tolerance response to ethanol. (**A**) In each block (day) of the experiment, the animal received ethanol injection after the presentation of the stimulus. (**B**) The change in the body temperature was measured 30, 60, 90, and 120 min after ethanol administration. Initially, the hypothermic effect of ethanol decreased the body temperature of animals. After several training days, however, animals learned to activate a tolerance response upon observing the stimulus, resulting in smaller deviations from the temperature setpoint. If the stimulus was not followed by ethanol injection, as in the first day of extinction (E1), the activation of the conditioned tolerance response resulted in an increase in body temperature. The tolerance response was weakened after several (four) extinction sessions, resulting in increased deviation from the setpoint in the first day of re-acquisition (R1), where presentation of the cue was again followed by ethanol injection.

first extinction trial on the ninth day where the ethanol was omitted, the animal's temperature exhibited a significant increase above normal after cue presentation. This indicates that the enhanced tolerance response to ethanol is triggered by the cue, and results in an increase of temperature in order to compensate for the forthcoming ethanol-induced hypothermia. Thus, this tolerance response is mediated by associative learning processes, and is aimed at regulating temperature. Here we demonstrate that the integration of HR and RL processes accounts for this phenomenon.

We simulate the model in an artificial environment where on every trial, the agent can choose between initiating a tolerance response and doing nothing, upon observing a cue (*Figure 3A*). The cue is then followed by a forced drop of temperature, simulating the effect of ethanol (*Figure 3B*). We also assume that in the absence of injection, the temperature does not change. However, if the agent chooses to initiate the tolerance response in this condition, the temperature increases gradually (*Figure 3D*). Thus, if ethanol injection is preceded by cue-triggered tolerance response, the combined effect (*Figure 3F*, as superposition of *Figure 3B,D*) will have less deviation from the setpoint as compared to when no response is taken (*Figure 3B*). As punishment (as the opposite of reward) in our model is defined by the extent to which the deviation from the setpoint increases, the 'null' response will have a bigger punishing value than the 'tolerance' response and thus, the agent gradually reinforces the 'tolerance' action (*Figure 3C*) (More precisely, the rewarding value of each action is defined by the sum of discounted drive-reductions during the 24 hr upon taking that action). This results in gradual fade of the ethanol-induced deviation of temperature from setpoint (*Figure 3E*; See *Figure 3—source data 1* for simulation details).

Clearly, if after this learning process cue-presentation is no longer followed by ethanol injection (as in the first extinction trial, E1), the cue-triggered tolerance response increases the temperate beyond the setpoint (*Figure 3E*).

In general, these results show that the tolerance response caused by predicted hypothermia is an optimal behavior in terms of minimizing homeostatic deviation and thus, maximizing reward. Thus, this optimal homeostatic maintenance policy is acquired by associative learning mechanisms.

Our theory implies that animals are capable of learning not only Pavlovian (e.g. shivering, or tolerance to ethanol), but also instrumental anticipatory responding (e.g., pressing a lever to receive warmth, in response to a cold-predicting cue). This prediction is in contrast to the theory of predictive homeostasis (also known as allostasis) where anticipatory behaviors are only *reflexive* responses to the predicted homeostatic deprivation upon observing cues (*Woods & Ramsay, 2007*; *Sterling, 2012*).

## Behavioral plausibility of drive: accounting for key phenomena

The definition of the drive function (*Equation 1*) in our model has two degrees of freedom: *m* and *n* are free parameters whose values determine the properties of the homeostatic space metric.

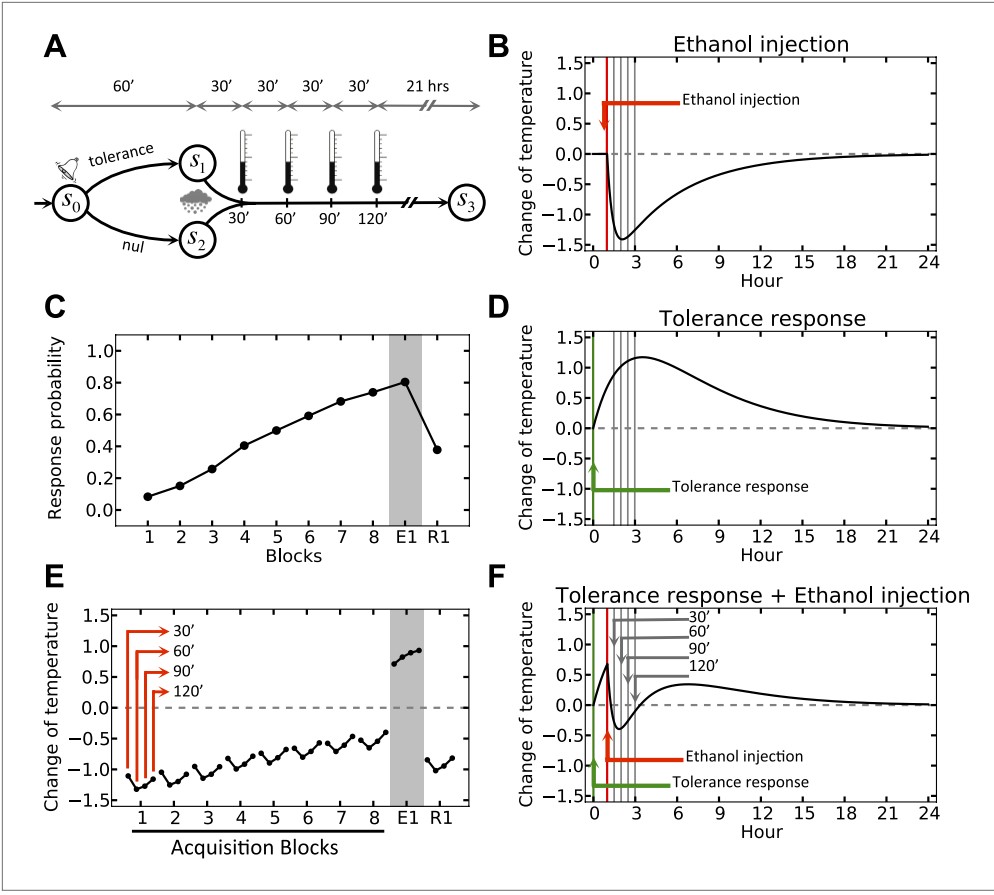

**Figure 3**. Simulation result on anticipatory responding. (**A**) In every trial, the simulated agent can choose between initiating a tolerance response and doing nothing, upon observing the stimulus. Regardless of the agent's choice, ethanol is administered after 1 hr, followed by four temperature measurements every 30 min. (**B**) Dynamics of temperature upon ethanol injection. (**C**) Learning curve for choosing the 'tolerance' response. (**D**) Dynamics of temperature upon initiating the tolerance response. (**E**) Temperature profile during several simulated trails. (**F**) Dynamics of temperature upon initiating the tolerance response, followed by ethanol administration. Plots c and e are averaged over 500 simulated agents.

The following source data is available for figure 3:

**Source data 1**. Free parameters for the anticipatory responding simulation.

Appropriate choice of *m* and *n* (*n* > *m* > 1) permits our theory to account for the following four key behavioral phenomena in a unified framework. First, it accounts for the fact that the reinforcing value of an appetitive outcome increases as a function of its dose ($K_t$) (**Figure 4A**):

$$\frac{\partial r(H_t, K_t)}{\partial k_{j,t}} > 0 \quad : \quad for\, K_t = \left(0, 0, \ldots, k_{j,t}, \ldots, 0\right) \text{ and } k_{j,t} > 0 \tag{6}$$

This is supported by the fact that in progressive ratio schedules of reinforcement rats maintain higher breakpoints when reinforced with bigger appetitive outcomes, reflecting higher motivation toward them (**Hodos, 1961**; **Skjoldager et al., 1993**). Secondly, the model accounts for the potentiating effect of the deprivation level on the reinforcing value (i.e., food will be more rewarding when the animal is hungrier) (**Figure 4B,C**):

$$\frac{\partial r(H_t, K_t)}{\partial \left|h_j^* - h_{j,t}\right|} > 0 \quad : \quad for\, K_t = \left(0, 0, \ldots, k_{j,t}, \ldots, 0\right) \text{ and } k_{j,t} > 0 \tag{7}$$

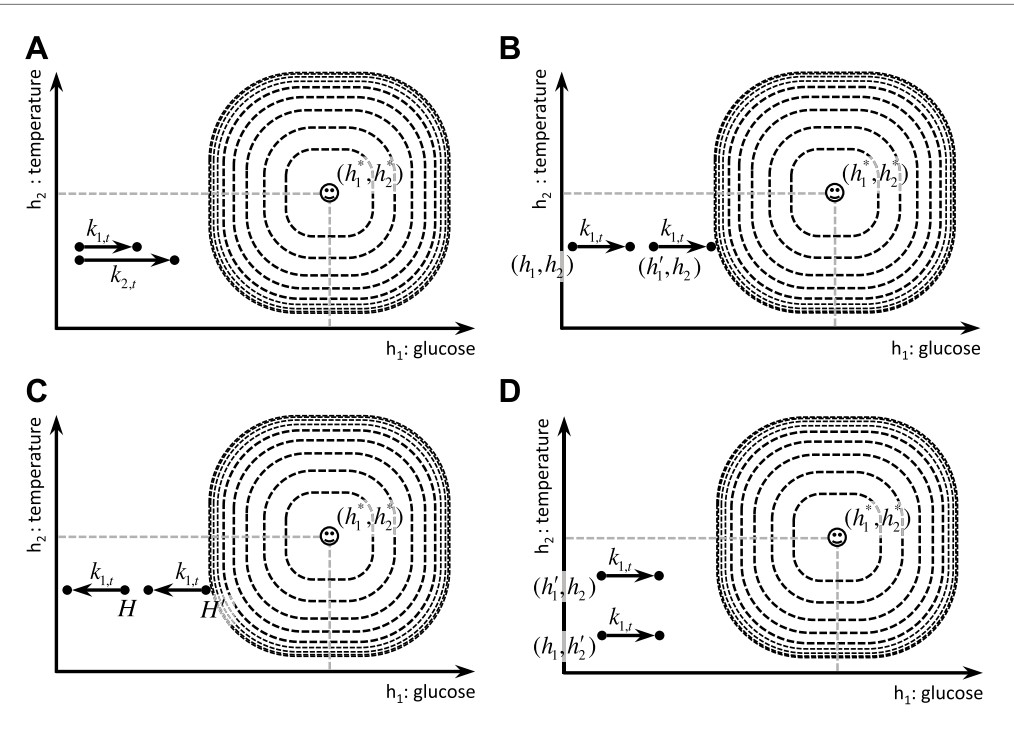

**Figure 4**. Schematic illustration of the behavioral properties of the drive function. (**A**) excitatory effect of the dose of outcome on its rewarding value. (**B**, **C**) excitatory effect of deprivation level on the rewarding value of outcomes: Increased deprivation increases the rewarding value of reducing drive (**B**), and increases the punishing value of increasing drive (**C**). (**D**) inhibitory effect of irrelevant drives on the rewarding value of outcomes.

This is consistent with experimental evidence showing that the level of food deprivation in rats increases the breakpoint in a progressive ratio schedule (**Hodos, 1961**). Note that this point effectively establishes a formal extension for the 'incentive' concept as defined by incentive salience theory (**Berridge, 2012**) (Discussed later).

Thirdly, the theory accounts for the inhibitory effect of irrelevant drives, which is consistent with a large body of behavioral experiments showing competition between different motivational systems (See **Dickinson & Balleine, 2002** for a review). In other words, as the deprivation level for one need increases, it inhibits the rewarding value of other outcomes that satisfy irrelevant motivational systems (**Figure 4D**):

$$\frac{\partial r(H_t, K_t)}{\partial \left| h_i^* - h_{i,t} \right|} > 0 \quad : \quad \text{for all } i \neq j, \text{ where } K_t = \left( 0, 0, \ldots, k_{j,t}, \ldots, 0 \right) \text{ and } k_{j,t} > 0 \qquad (8)$$

Intuitively, one does not play chess, or even search for sex, on an empty stomach. As some examples, calcium deprivation reduces the appetite for phosphorus, and hunger inhibits sexual behavior (**Dickinson & Balleine, 2002**).

Finally, the theory naturally captures the risk-aversive nature of behavior. The rewarding value in our model is a concave function of the corresponding outcome magnitude:

$$\frac{\partial^2 r(H_t, K_t)}{\partial k_{j,t}^2} < 0 \quad : \quad \text{for } K_t = \left( 0, 0, \ldots, k_{j,t}, \ldots, 0 \right) \text{ and } k_{j,t} > 0 \qquad (9)$$

It is well known that the concavity of the economic utility function is equivalent to risk aversion (**Mas-Colell et al., 1995**). Indeed, simulating the model shows that when faced with two options with equal expected payoffs, the model learns to choose the more certain option as opposed to the risky one (**Figure 5**; See **Figure 5—source data 1** for simulation details). This is because frequent small deviations from the setpoint are preferable to rare drastic deviations. In fact, our theory suggests the

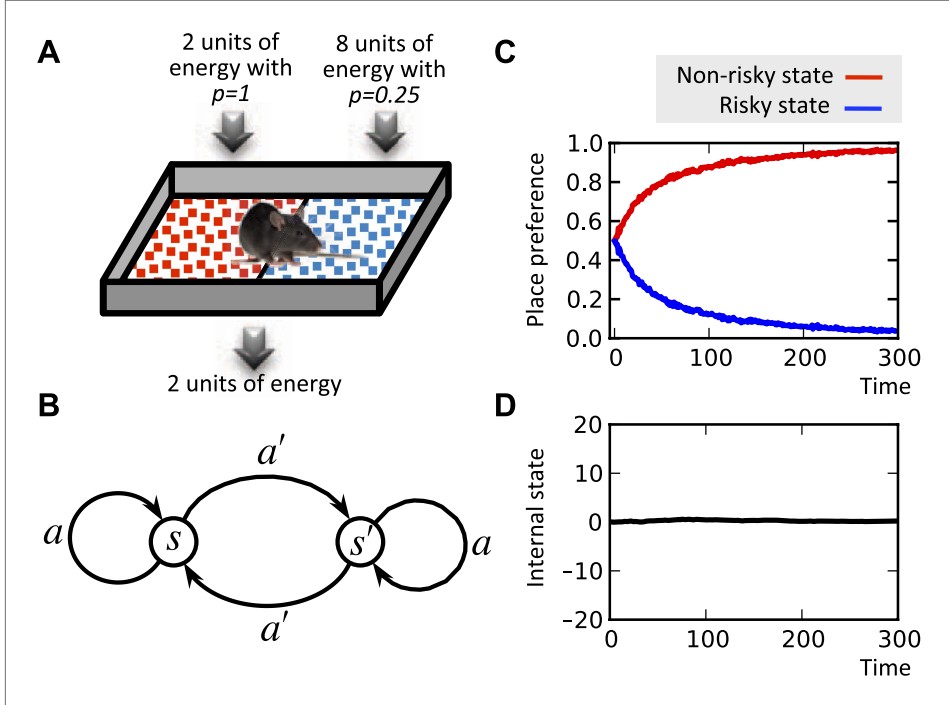

**Figure 5**. Risk aversion simulation. In a conditioned place preference paradigm, the agent's presence in the left and the right compartments has equal expected payoffs, but different levels of risk (**A**) Panel (**B**) shows the Markov decision process of the same task. In fact, in every trial, the agent chooses whether to stay it the current compartment, or transit to the other one. The average input of energy per trial, regardless of the animal's choice, is set such that it is equal to the animal's normal energy expenditure. Thus, the internal state stays close to its initial level, which is equal to the setpoint here (**D**). The model learns to prefer the non-risky over the risky compartments (**C**) in order to avoid severe deviations from the setpoint.

The following source data is available for figure 5:

**Source data 1**. Free parameters for the risk-aversion simulations.

intuition that when the expected physiological instability caused by two behavioral options are equal, organisms do not choose the risky option, because the severe, though unlikely, physiological instabilities that it can cause might be life-threatening.

Our unified explanation for the above four behavioral patterns suggests that they may all arise from the functional form of the mapping from the physiological to the motivational state. In this sense, we propose that these behavioral phenomena are signatures of the coupling between the homeostatic and the associative learning systems. We will discuss later that $m$, $n$, and $H^*$ can be regarded as free parameters of an evolutionary process, which eventually determine the equilibrium density of the species.

Note that the equations in this section hold only when the internal state remains below the setpoint. However, the drive function is symmetric with respect to the setpoint and thus, analogous conclusions can be derived for other three quarters of the homeostatic space.

## Stepping back from the brink

Since learning requires experience, learning whether an action in a certain internal state decreases or increases the drive (i.e. is rewarding or punishing, respectively) would require our model to have experienced that internal state. Living organisms, however, cannot just experience internal states with extreme and life threatening homeostatic deviations in order to learn that the actions that cause them are bad. For example, once the body temperature goes beyond 45°C, the organism can never return.

We now show how our model manages this problem; that is, it avoids voluntarily experiencing extreme homeostatic deviations and hence ensures that the animal does not voluntarily endanger its physiological integrity (simulations in *Figure 6*). In the simplest case, let us assume that the model is

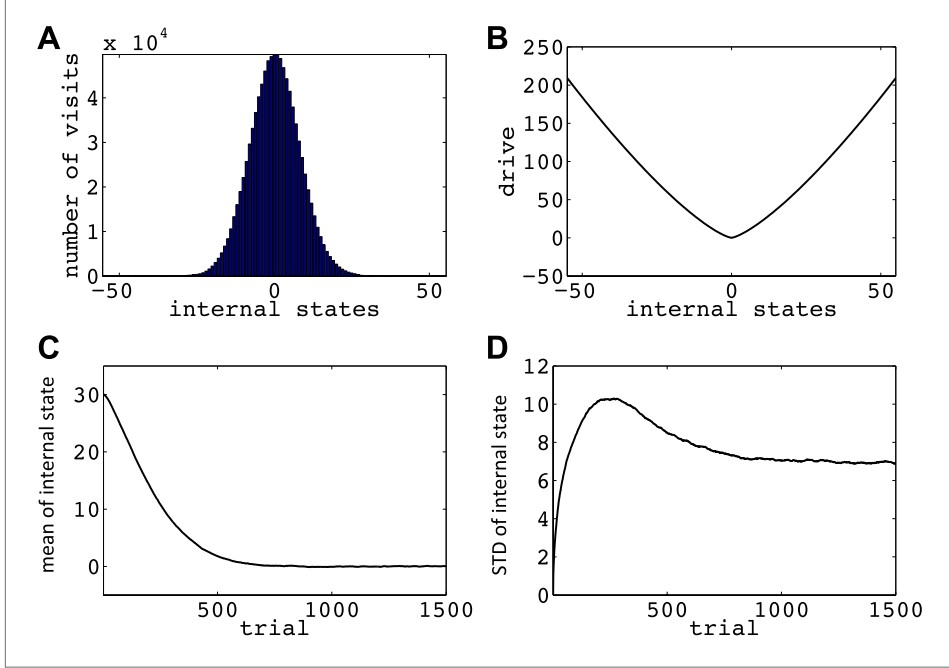

**Figure 6**. Simulations showing that the model avoids extreme deviations. Starting from 30, the agent can either decrease or increase its internal state by one unit in each trial. (**A**) The number of visits at each internal state after $10^6$ trials. (**B**) The drive function in the one-dimensional homeostatic space. (setpoint = 0). The mean (**C**) and standard deviation (**D**) of the internal state of $10^5$ agents, along 1500 trials.

The following source data and figure supplements are available for figure 6:

**Source data 1**. Free parameters for the simulations showing that the model avoids extreme homeostatic deviations.

**Figure supplement 1**. The Markov Decision Process used for simulation results presented in *Figure 6* and *Figure 6—figure supplements 2–7*.

**Figure supplement 2**. Value function (**A**) and choice preferences (**B**) for state–action pairs after simulating one agent for $10^6$ trials (As in *Figure 6*).

**Figure supplement 3**. Simulation results replicating *Figure 6*, with the difference that the initial internal state was zero.

**Figure supplement 4**. Simulation results replicating *Figure 6*, with the difference that the initial internal state was zero, and the rate of exploration, *β*, was 0.03.

**Figure supplement 5**. Simulation results replicating *Figure 6*, with the difference that the initial internal state was zero, and also *m* = *n* = 1.

**Figure supplement 6**. Simulation results replicating *Figure 6*, with the difference that the initial internal state was zero, and the discount factor, *γ*, was zero.

**Figure supplement 7**. Simulation results replicating *Figure 6*, with the difference that the initial internal state was zero, and the discount factor, *γ*, was one (no discounting).

tabula rasa: it starts from absolute ignorance about the value of state–action pairs, and can freely change its internal state in the homeostatic space. In a one-dimensional space, it means that the agent can freely increase or decrease the internal state (*Figure 6—figure supplement 1*). As the value of 'increase' and 'decrease' actions at all internal states are initialized to zero, the agent starts by performing a random walk in the homeostatic space. However, the probability of choosing the same action for $z$ times in a row decreases exponentially as $z$ increases ($p(z) = 2^{-z}$): for example, the probability of

choosing 'increase' is $2^{-1} = 0.5$, the probability of choosing two successive 'increases' is $2^{-1} = 0.25$, the probability of choosing three successive 'increases' is $2^{-3} = 0.125$, and so on. Thus, it is highly likely for the agent to return at least one step back, before getting too far from its starting point. When the agent returns to a state it had previously experienced, going in the same deviation-increasing direction will be less likely than the first time (i.e., than 50–50), since the agent has already experienced the punishment caused by that state–action pair once. Repetition of this process results in the agent gradually getting more and more attracted to the setpoint, without ever having experienced internal states that are beyond a certain limit (i.e. the brink of death).

Simulating the model in a one-dimensional space shows that even after starting from a rather deviated internal state (initial state = 30, setpoint = 0), the agent never visits states with a deviation of more than 40 units after $10^6$ trials (every action is assumed to change the state by one unit) (**Figure 6A**; See **Figure 6—figure supplements 1,2**, and **Figure 6—source data 1** for simulation details). Also, simulating $10^5$ agents over 1500 trials (starting from state 30) shows that the mean value of the internal state across all agents converges to the setpoint (**Figure 5C**), and its variance converges to a steady-state level (**Figure 5D**). This shows that all agents stay within certain bounds around the setpoint (The maximum deviation from the setpoint among all the $10^5$ agents over the 1500 trials was 61). Also, this property of the model is shown to be insensitive to the parameters of the model, like the initial internal state (**Figure 6—figure supplement 3**), the rate of exploration (**Figure 6—figure supplement 4**), $m$ and $n$ (**Figure 6—figure supplement 5**), or the discount factor (**Figure 6—figure supplements 6,7**). These parameters only affect the rate of convergence or the distribution over visited states, but not the general property of never-visiting-drastic-deviations (existence of a boundary). Moreover, this property can be generalized to multi-dimensional homeostatic spaces. Therefore, our theory suggests a potential normative explanation for how animals (who might be a priori naïve about potential dangers of certain internal states) would learn to avoid extreme physiological instability, without ever exploring how good or bad they are.

## Orosensory-based approximation of post-ingestive effects

As mentioned, we hypothesize that orosensory properties of food and water provide the animal with an estimate, $\hat{K}_t$, of their true post-ingestive effect, $K_t$, on the internal state. Such association between sensory and post-ingestive properties could have been developed through prior learning (**Swithers et al., 2009**; **Swithers et al., 2010**; **Beeler et al., 2012**) or evolutionary mechanisms (**Breslin, 2013**). Based on this sensory approximation, the only information required to compute the reward (and thus the reward prediction error) is the current physiological state ($H_t$) and the sensory-based approximation of the nutritional content of the outcome ($\hat{K}_t$):

$$r\left(H_t, \hat{K}_t\right) = D\left(H_t\right) - D\left(H_t + \hat{K}_t\right) \tag{10}$$

Clearly, the evolution of the internal state itself depends only on the actual ($K_t$) post-ingestive effects of the outcome. That is $H_{t+1} = H_t + K_t$.

According to **Equation 10**, the reinforcing value of food and water outcomes can be approximated as soon as they are sensed/consumed, without having to wait for the outcome to be digested and the drive to be reduced. This proposition is compatible with the fact that dopamine neurons exhibit instantaneous, rather than delayed, burst activity in response to unexpected food reward (**Schneider, 1989**; **Schultz et al., 1997**). Moreover, it might provide a formal explanations for the experimental fact that intravenous injection (and even intragastric intubation, in some cases) of food is not rewarding even though its drive reduction effect is equal to when it is ingested orally (**Miller & Kessen, 1952**) (See also **Ren et al., 2010**). In fact, if the post-ingestive effect of food is estimated by its sensory properties, the reinforcing value of intravenously injected food that lacks sensory aspects will be effectively zero. In the same line of reasoning, the theory suggests that animals' motivation toward palatable foods, such as saccharine, that have no caloric content (and thus no need-reduction effect) is due to erroneous overestimation of their drive-reduction capacity, misguided by their taste or smell. Note that the rationality of our theory, as shown in **Equation 5**, holds only as long as $\hat{K}_t$ is an unbiased estimation of $K_t$. Otherwise, pathological conditions could emerge.

Last but not least, the orosensory-based approximation provides a computational hypothesis for the separation of reinforcement and satiation effects. A seminal series of experiments (**McFarland, 1969**) demonstrated that the reinforcing and satiating (i.e., need reduction) effects of drinking behavior, dissociable from one another, are governed by the orosensory and alimentary components of the water, respectively.

Two groups of water-deprived animals learned to press a green key to self-administer water orally. After this pre-training session, pressing the green key had no consequence anymore, whereas pressing a novel yellow key resulted in the oral delivery of water in one group, and intragastric (through a fistula) delivery of water in the second group. Results showed that the green key gradually extinguished in both groups (*Figure 7A,B*). During this time, responding on the yellow key in the oral group initially increased but then gradually extinguished (rise-fall pattern; *Figure 7A*). The second group, however, showed no motivation for the yellow key (*Figure 7B*). This shows that only oral, but not intragastric, self-administration of water is reinforcing for thirsty animals. Our model accounts for these behavioral dynamics.

Simulating the model shows that the agent's subjective probability of receiving water upon pressing the green key gradually decreases to zero in both groups (*Figure 8C,D*). As this predicted outcome (alimentary content) decreases, its approximated thirst-reduction effect (equal to reward in our framework) decreases as well, resulting in the extinction of pressing the green key (*Figure 8A,B*). As for the yellow key, the oral agent initially increases the rate of responding (*Figure 8A*) as the subjective probability of receiving water upon pressing the yellow key increases (*Figure 8C*). Gradually, however, the internal state of the animal reaches the homeostatic setpoint (*Figure 8E*), resulting in diminishing motivation (thirst-reduction effect) of seeking water (*Figure 8A*). Thus, our model shows that whereas the ascending limb of the response curve represents a learning effect, the descending limb is due to mitigated homeostatic imbalance (i.e., unlearning vs. satiation). Notably, classical RL models only explain the ascending, and classical HR models only explain the descending pattern.

In contrast to the oral agent, the fistula agent never learns to press the yellow key (*Figure 8B*). This is because the approximated alimentary content attributed to this response remains zero (*Figure 8D*) and so does its drive-reduction effect. Note that as above, the sensory-based approximation ($\hat{K}_t$) of the alimentary effect of water in the oral and fistula cases is assumed to be equal to its actual effect ($K_t$) and zero, respectively (See *Figure 8—figure supplements 1,2*, and *Figure 8—source data 1* for simulation details).

Our theory also suggests that in contrast to reinforcement (above), satiation is independent of the sensory aspects of water and only depends on its post-ingestive effects. In fact, experiments show that when different proportions of water were delivered via the two routes in different groups, satiation (i.e., suppression of responding) only depended on the total amount of water ingested, regardless of the delivery route (*McFarland, 1969*).

Our model accounts for these data (*Figure 9*), since the evolution of the internal state only depends on the actual water ingested. For example, whether water is administered completely orally (*Figure 9*, left column) or half-orally-half-intragastrically (*Figure 9*, right column), the agent stops seeking water when the setpoint is reached. As only oral delivery is sensed, the subjective outcome magnitude converges to 1 (*Figure 9C*) and 0.5 (*Figure 9D*) units for the two cases, respectively. When the setpoint is reached, consuming more water results in overshooting the setpoint (increasing homeostatic

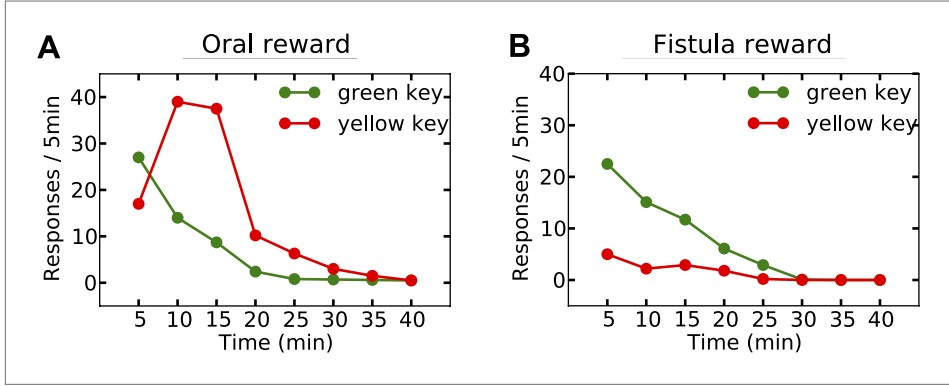

**Figure 7**. Experimental results (adapted from *McFarland, 1969*) on learning the reinforcing effect of oral vs. intragastric delivery of water. Thirsty animals were initially trained to peck at a green key to receive water orally. In the next phase, pecking at the green key had no consequence, while pecking at a novel yellow key resulted in oral delivery of water in one group (**A**), and intragastric injection of the same amount of water through a fistula in a second group (**B**). In the first group, responding was rapidly transferred from the green to the yellow key, and then suppressed. In the fistula group, the yellow key was not reinforced.

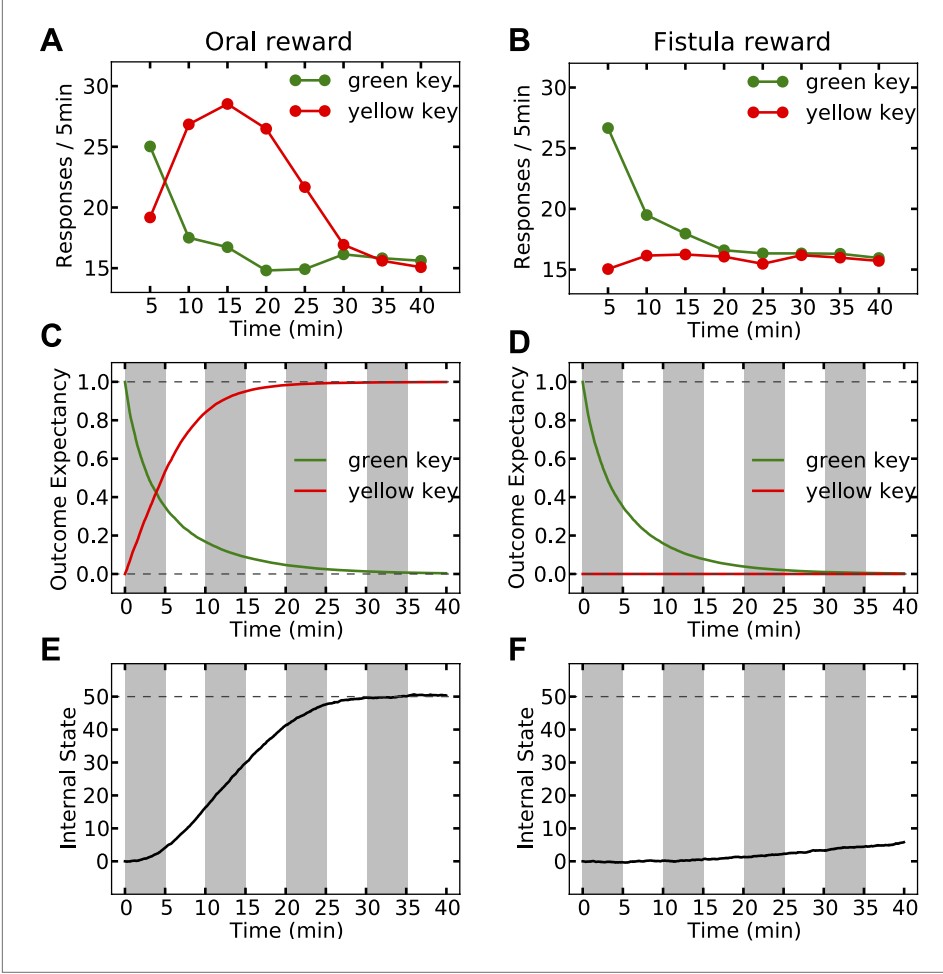

**Figure 8**. Simulation results replicating the data from *McFarland (1969)* on learning the reinforcing effect of oral vs. intragastric delivery of water. As in the experiment, two groups of simulated agents were pre-trained to respond on the green key to receive oral delivery of water. During the test phase, the green key had no consequence, whereas a novel yellow key resulted in oral delivery in one group (**A**) and intragastric injection in the second group (**B**). All agents started this phase in a thirsty state (initial internal state = 0; setpoint = 0). In the oral group, responding transferred rapidly from the green to the yellow key and was then suppressed (**A**) as the internal state approached the setpoint (**E**). This transfer is due to gradually updating the subjective probability of receiving water outcome upon responding on either key (**C**). In the fistula group, as the water was not sensed, the outcome expectation converged to zero for both keys (**D**) and thus, responding was extinguished (**B**). As a result, the internal state changed only slightly (**F**).

The following source data and figure supplements are available for figure 8:

**Source data 1**. Free parameters for the reinforcing vs. satiation simulations.

**Figure supplement 1**. A model-based homeostatic RL system. Upon performing an action in a certain state, the agent receives an outcome, $K_t$, which results in the internal state to shift from $H_t$ to $H_t + K_t$.

**Figure supplement 2**. The Markov Decision Process used for simulating the reinforcing vs. satiation effects of water.

deviation) and thus, is punishing. Therefore, both agents self-administer the same total amount of water, equal to what is required for reaching the setpoint.

However, as the sensed amount of water is bigger in the completely-oral case, water-seeking behavior is approximated to have a higher thirst-reduction effect. As a result, the reinforcing value of water-seeking is higher in the oral case (as compared to the half-oral-half-intragastric case) and thus, the rate of responding is higher. This, in turn, results in faster convergence of the internal state to the setpoint

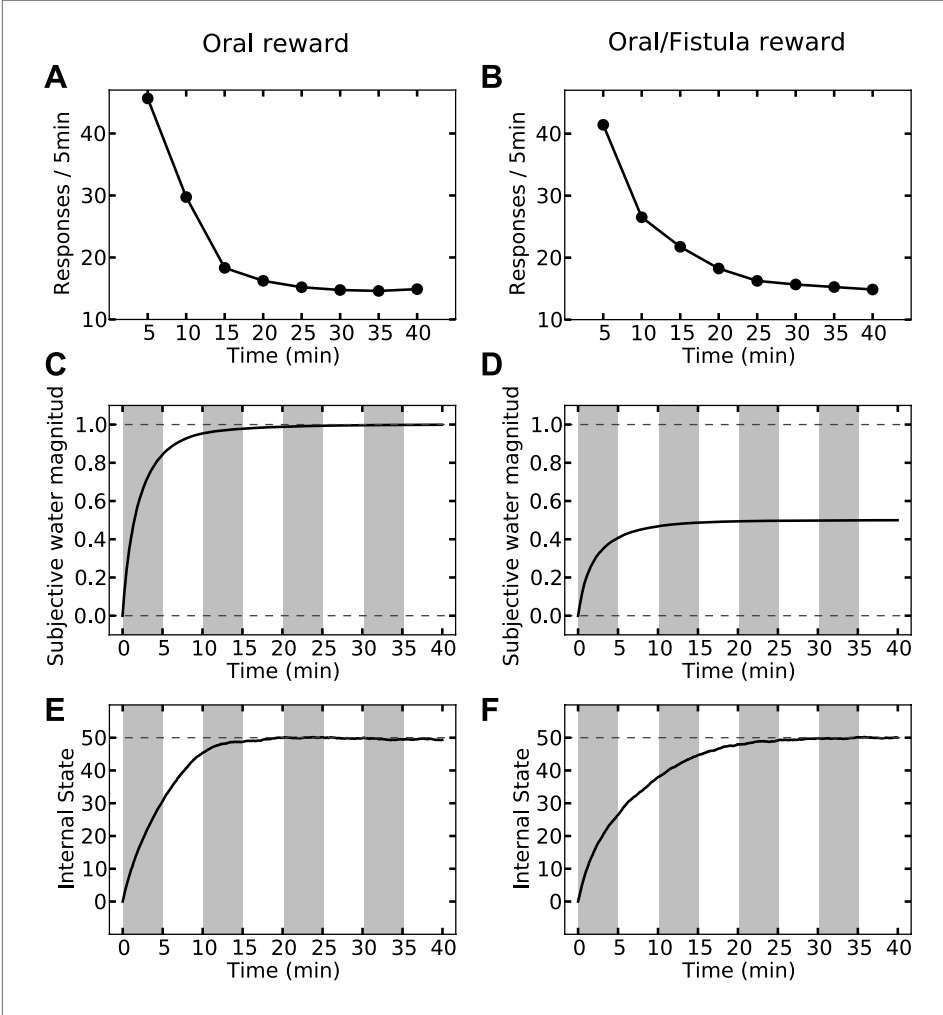

**Figure 9**. Simulation results of the satiation test. Left column shows results for the case where water was received only orally. Rate of responding drops rapidly (**A**) as the internal state approaches the setpoint (**E**). Also, the agent learns rapidly that upon every key pecking, it receives 1.0 unit of water (**C**). On the right column, upon every key-peck, 0.5 unit of water is received orally, and 0.5 unit is received via the fistula. As only oral delivery is sensed by the agent, the subjective outcome-magnitude converges to 0.5 (**D**). As a result, the reinforcing value of key-pecking is less than that of the oral case and thus, the rate of responding is lower (**B**). This in turn results in slower convergence of the internal state to the setpoint (**F**). The MDP and the free parameters used for simulation are the same as in *Figure 8*.

(compare *Figure 9E,F*). In this respect, we predict that the oral/fistula proportion affects the speed of satiation: the higher the proportion is, the faster the satiety state is reached and thus, the faster the descending limb of responding emerges.

## Discussion

Theories of conditioning are founded on the argument that animals seek reward, while reward may be defined, at least in the behaviorist approach, as what animals seek. This apparently circular argument relies on the hypothetical and out-of-reach axiom of reward-maximization as the behavioral objective of animals. Physiological stability, however, is an observable fact. Here, we develop a coherent mathematical theory where physiological stability is put as the basic axiom, and reward is defined in physiological terms. We demonstrated that reinforcement learning algorithms under such a definition of physiological reward lead to optimal policies that both maximize reward collection and minimize homeostatic needs. This argues for behavioral rationality of physiological integrity maintenance and

further shows that temporal discounting of rewards is paramount for homeostatic maintenance. Furthermore, we demonstrated that such integration of the two systems can account for several behavioral phenomena, including anticipatory responding, the rise-fall pattern of food-seeking response, risk-aversion, and competition between motivational systems. Here we argue that our framework may also shed light on the computational role of the interaction between the brain reward circuitry and the homeostatic regulation system; namely, the modulation of midbrain dopaminergic activity by hypothalamic signals.

## Neural substrates

Homeostatic regulation critically depends on sensing the internal state. In the case of energy regulation, for example, the arcuate nucleus of the hypothalamus integrates peripheral hormones including leptin, insulin, and ghrelin, whose circulating levels reflect the internal abundance of fat, abundance of carbohydrate, and hunger, respectively (*Williams & Elmquist, 2012*). In our model, the deprivation level has an excitatory effect on the rewarding value of outcomes (*Equation 7*) and thus on the reward prediction error (RPE). Consistently, recent evidence indicates neuronal pathways through which energy state-monitoring peptides modulate the activity of midbrain dopamine neurons, which supposedly carry the RPE signal (*Palmiter, 2007*).

Namely, orexin neurons, which project from the lateral hypothalamus area to several brain regions including the ventral tegmental area (VTA) (*Sakurai et al., 1998*), have been shown to have an excitatory effect on dopaminergic activity (*Korotkova et al., 2003*; *Narita et al., 2006*), as well as feeding behavior (*Rodgers et al., 2001*). Orexin neurons are responsive to peripheral metabolic signals as well as to the animal's deprivation level (*Burdakov et al., 2005*), as they are innervated by orexigenic and anorexigenic neural populations in the arcuate nucleus where circulating peptides are sensed. Accordingly, orexin neurons are suggested to act as an interface between internal states and the reward learning circuit (*Palmiter, 2007*). In parallel with the orexinergic pathway, ghrelin, leptin and insulin receptors are also expressed on the VTA dopamine neurons, providing a further direct interface between the HR and RL systems. Consistently, whereas leptin and insulin inhibit dopamine activity and feeding behavior, ghrelin has an excitatory effect on them (See *Palmiter, 2007* for a review).

The reinforcing value of food outcome (and thus RPE signal) in our theory is not only modulated by the internal state, but also by the orosensory information that approximates the need-reduction effects. In this respect, endogenous opioids and $\mu$-opioid receptors have long been implicated in the hedonic aspects of food, signaled by its orosensory properties. Systemic administration of opioid antagonists decreases subjective pleasantness rating and affective responses for palatable foods in humans (*Yeomans & Wright, 1991*) and rats (*Doyle et al., 1993*), respectively. Supposedly through modulating palatability, opioids also control food intake (*Sanger & McCarthy, 1980*) as well as instrumental food-seeking behavior (*Cleary et al., 1996*). For example, opioid antagonists decrease the breakpoint in progressive ratio schedules of reinforcement with food (*Barbano et al., 2009*), whereas opioid agonists produce the opposite effect (*Solinas & Goldberg, 2005*). This reflects the influence of orosensory information on the reinforcing effect of food. Consistent with our model, these influences have mainly been attributed to the effect of opiates on increasing extracellular dopamine levels in the Nucleus Accumbens (NAc) (*Devine et al., 1993*) through its action on $\mu$-opioid receptors in the VTA and NAc (*Noel & Wise, 1993*; *Zhang & Kelley, 1997*).

Such orosensory-based approximation of nutritional content, as discussed before, could have been obtained through evolutionary processes (*Breslin, 2013*), as well as through prior learning (*Beeler et al., 2012*; *Swithers et al., 2009, 2010*). In the latter case, approximations based on orosensory or contextual cues can be updated so as to match the true nutritional value, resulting in a rational neural/behavioral response to food stimuli (*de Araujo et al., 2008*).

## Irrational behavior: the case of over-eating

Above, we developed a normative theory for reward-seeking behaviors that lead to homeostatic stability. However, animals do not always follow rational behavioral patterns, notably as exemplified in eating disorders, drug addiction, and many other psychiatric diseases. Here we discuss one prominent example of such irrational behavior within the context of our theory.

Binge eating is a disorder characterized by compulsive eating even when the person is not hungry. Among the many risk factors of developing binge eating, a prominent one is having easy access to hyperpalatable foods, commonly defined as those loaded with fat, sugar, or salt (*Rolls, 2007*). As an

attempt to explain this risk factor, we discuss one of the points of vulnerability of our theory that can induce irrational choices and thus, pathological conditions.

Over-seeking of hyperpalatable foods is suggested to be caused by motivational systems escaping homeostatic constraints, supposedly as a result of the inability of internal satiety signals in blocking the opioid-based stimulation of DA neurons (*M. Zhang & Kelley, 2000*). Stimulation of *μ*-opioid receptors in the NAc, for example, is demonstrated to preferentially increase the intake of high-fat food (*Glass et al., 1996*; *Zhang & Kelley, 2000*), and hyperpalatable foods are shown to trigger potent release of DA into the NAc (*Nestler, 2001*). Moreover, stimulation of the brain reward circuitry (*Will et al., 2006*), as well as DA receptor agonists (*Cornelius et al., 2010*) are shown to induce hedonic overeating long after energy requirements are met, suggesting the hyper-palatability factor to be drive-independent.

Motivated by these neurobiological findings, one way to formulate the overriding of the homeostatic satiety signals by hyperpalatable foods is to assume that the drive-reduction reward for these outcomes is augmented by a drive-independent term, $T$ ($T > 0$ for palatable foods, and $T = 0$ for 'normal' foods):

$$r\left(H_t, K_t\right) = D\left(H_t\right) - D\left(H_t + K_t\right) + T$$

(11)

In other words, even when the setpoint is reached and thus, the drive-reduction effect of food is zero or even negative, the term $T$ overrides this signal and results in further motivation for eating (See 'Materials and methods' for alternative formulations of *Equation 11*). Simulating this hypothesis shows that when a deprived agent (initial internal state = −50) is given access to normal food, the internal state converges to the setpoint (*Figure 10C*). When hyperpalatable food with equal caloric content ($K$ is the same for both types of food) is made available instead, the steady level of the internal state goes beyond the setpoint (*Figure 10C*). Moreover, the total consumption of food is higher in the latter case (*Figure 8D*), reflecting overeating. In fact, the inflated hedonic aspect of the hyperpalatable food causes it to be sought and consumed to a certain extent, even after metabolic demands are fulfilled. One might speculate that such persistent overshoot would result in excess energy storage, potentially leading to obesity.

Simulating the model in another condition where the agent has 'concurrent' access to both types of foods shows significant preference of the hyperpalatable food over the normal food (*Figure 10E*), and the internal state again converges to a higher-than-setpoint level (*Figure 10F*). This is in agreement with the evidence showing that animals strongly prefer highly palatable to less palatable foods (*McCrory et al., 2002*). (See *Figure 10—source data 1* for simulation details)

## Relationship to classical drive-reduction theory

Our model is inspired by the drive reduction theory of motivation, initially proposed by Clark Hull (*Hull, 1943*), which became the dominant theory of motivation in psychology during the 1940s and 1950s. However, major criticisms have been leveled against this theory over the years (*McFarland, 1969*; *Savage, 2000*; *Berridge, 2004*; *Speakman et al., 2011*). Here we propose that our formal theory alleviates some of major faults of the classical drive-reduction. Firstly, the classical drive-reduction does not explain anticipatory responding in which animals paradoxically voluntarily increase (rather than decrease) their drive deviation, even in the absence of any physiological deficit. As we demonstrated, such apparently maladaptive responses are optimal in terms of both reward-seeking and ensuring physiological stability, and are thus acquired by animals.

Secondly, the drive reduction could not explain how secondary reinforcers (e.g., money, or a light that predicts food) gain motivational value, since they do not reduce the drive per se. Because our framework integrates an RL module with the HR reward computation, the drive reduction-induced reward of primary reinforcers can be readily transferred through the learning process to secondary reinforcers that predict them (i.e., Pavlovian conditioning) as well as to behavioral policies that lead to them (i.e., instrumental conditioning).

Finally, the original Hull's theory is in contradiction with the fact that intravenous injection of food is not rewarding, despite its drive-reduction effect. As we showed, this could be due to the orosensory-based approximation mechanism required for computing the reward.

Despite its limitations (discussed later), we would suggest that our modern re-formulation of the drive-reduction theory subject to specific assumptions (i.e., orosensory approximation, connection to RL, drive form) can serve as a framework to understand the interaction between internal states and motivated behaviors.

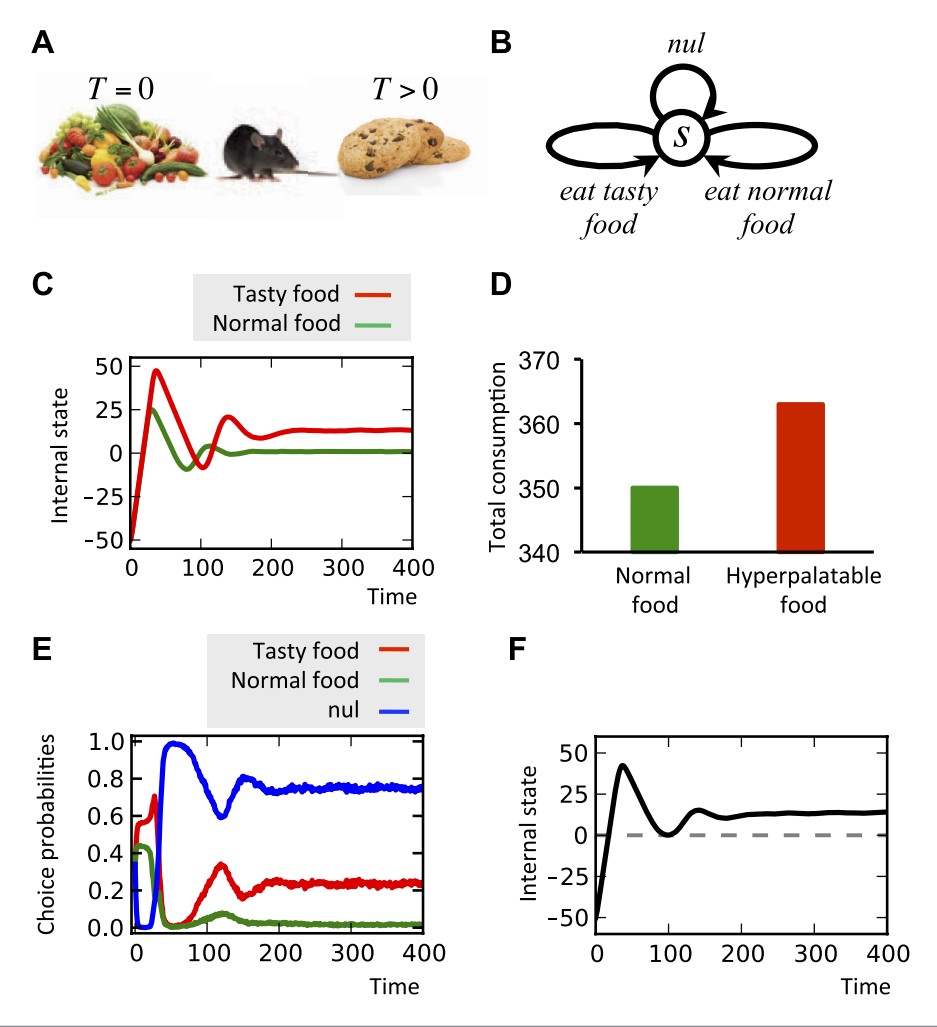

**Figure 10**. Simulating over-eating of hyperpalatable vs. normal food. (**A**) The simulated agent can consume normal ($T = 0$) or hyperpalatable ($T > 0$) food. The nutritional content, $K$, of both foods are equal. In the single-option task (**C**, **D**), one group of animals can only choose between normal food and nothing (nul), whereas the other group can choose between hyperpalatable food and nothing. Starting the task in a deprived state (initial internal state=$-50$), the internal state of the second, but not the first, group converges to a level above the setpoint (**C**) and the total consumption of food is higher in this group (**D**). In the multiple-choice task, the agents can choose between normal food, hyperpalatable food, and nothing (**B**). Results show that the hyperpalatable food is preferred over the normal food (**E**) and the internal state is defended at a level beyond the setpoint (**F**). See *Figure 10—source data 1* for simulation details.

The following source data is available for figure 10:

**Source data 1**. Free parameters for the over-eating simulations.

## Relationship to other theoretical models

Several previous RL-based models have also tried to incorporate the internal state into the computation of reward by proposing that reward increases as a linear function of deprivation level. That is, $r = w\bar{r}$, where $\bar{r}$ is a constant and $w$ is proportional to the deprivation level.

Interestingly, a linear approximation of our proposed drive-reduction reward is equivalent to assuming that the rewarding value of outcomes is equal to the multiplication of the deprivation level and the magnitude of the outcome. In fact, by rewriting *Equation 2* for the continuous case we will have:

$$r\left(H_t, K_t\right) \equiv \frac{dD(H_t + K_t)}{dK_t} \tag{12}$$

Using Taylor expansion, this reward can be approximated by:

$$r\left(H_t, K_t\right) \cong -K_t . \nabla D_H\left(H_t\right) + O\left(\nabla^2 D_H\left(H_t\right)\right) \tag{13}$$

Where $\nabla$ is the gradient operator, and $\nabla^2$ is the Laplace operator. Thus, a linear approximation of our proposed drive-reduction reward is equivalent to assuming that the rewarding value of outcomes is linearly proportional to their need-reduction capacity ($K_t$), as well as a function (the gradient of drive) of the deprivation level. In this respect, our framework generalizes and provides a normative basis to multiplicative forms of deprivation-modulated reward (e.g., decision field theory (*Busemeyer et al., 2002*), intrinsically motivated RL theory (*Singh et al., 2010*), and MOTIVATOR theory (*Dranias et al., 2008*)), where reward increases as a linear function of deprivation level. Moreover, those previous models cannot account for the non-linearities arising from our model; that is the inhibitory effect of irrelevant drives and risk aversion.

Whether the brain implements a nonlinear drive-reduction reward (as in *Equation 2*) or a linear approximation of it (as in *Equation 13*) can be examined experimentally. Assuming that an animal is in a slightly deprived state (*Figure 11A*), a linear model predicts that as the magnitude of the outcome increases, its rewarding value will increase linearly (*Figure 11B*). A non-linear reward, however, predicts an inverted U-shaped economic utility function (*Figure 11B*). That is, the rewarding value of a large outcome can be negative, if it results in overshooting the setpoint.

A more recent framework that also uses a multiplicative form of deprivation-modulated reward is the incentive salience theory (*Berridge, 2012*; *Zhang et al., 2009*). However, in contrast to the previous models and our framework, this model assumes that the rewarding value of outcomes and conditioned stimuli is learned as if the animal is in a reference internal state ($\psi = 1$). Let's denote this reward by $r(s, \psi = 1)$ for state $s$. At the time of encountering state $s$ in the future, the animal uses a factor, $\psi_t$, related to its current internal state, to modulate the real-time motivation of the animal: $r\left(s, \psi_t\right) = \psi_t . r(s, \psi = 1)$. In the case of conditioned tolerance to hypothermic agents, however, heat-producing response is motivated at the time of cue presentation, when the hypothermic agent is not administered yet. At this time, the animal's internal state is not deviated and thus, the motivational element, $\psi_t$, in the incentive salience theory does not provoke the tolerance response. Therefore, in our reading and unlike our framework, the incentive salience theory cannot give a computational account of anticipatory responding.

Another approach to integrate responsiveness to both internal and external states appeals to approximate inference techniques from statistical physics. The free energy theory of brain (*Friston, 2010*) proposes that organisms optimize their actions in order to minimize 'surprise'. Surprise is an information-theoretic notion measuring how inconceivable it is to the organism to find itself in a certain state. Assume that evolutionary pressure has compelled a species to occupy a restricted set of internal states, and $p(H_t)$ indicates the probability of occupying state $H_t$, after the evolution of admissible states has converged to an equilibrium density. Surprise is defined as the negative log-probability of $H_t$ occurring; $-\ln p(H_t)$.

We propose that our notion of drive is equivalent to surprise as utilized in the free energy (*Friston, 2010*) and interoceptive inference (*Seth, 2013*) frameworks. In fact, we propose that an organism has an equilibrium density, $p(.)$, with the following functional form:

$$p\left(H_t\right) \propto e^{-D(H_t)} = e^{-\sqrt[m]{\sum_{i=1}^{N}\left|h_i^* - h_{i,t}\right|^n}} \tag{14}$$

In order to stay faithful to this probability density (and ensure the survival of genes by remaining within physiological bounds), the organism minimizes surprise, which is equal to $-\ln p\left(H_t\right) = \sqrt[m]{\sum_{i=1}^{N}\left|h_i^* - h_{i,t}\right|^n}$. This specific form of surprise is equivalent to our definition of drive (*Equation 1*). The equivalency of reward maximization and physiological stability objectives in our model (*Equation 5*) shows that optimizing either homeostasis or sum of discounted rewards corresponds to prescribing a principle of least action applied to the surprise function.

Although our homeostatic RL and the free-energy theory are similar in spirit, several major differences can be mentioned. Most importantly, the two frameworks should be understood at different

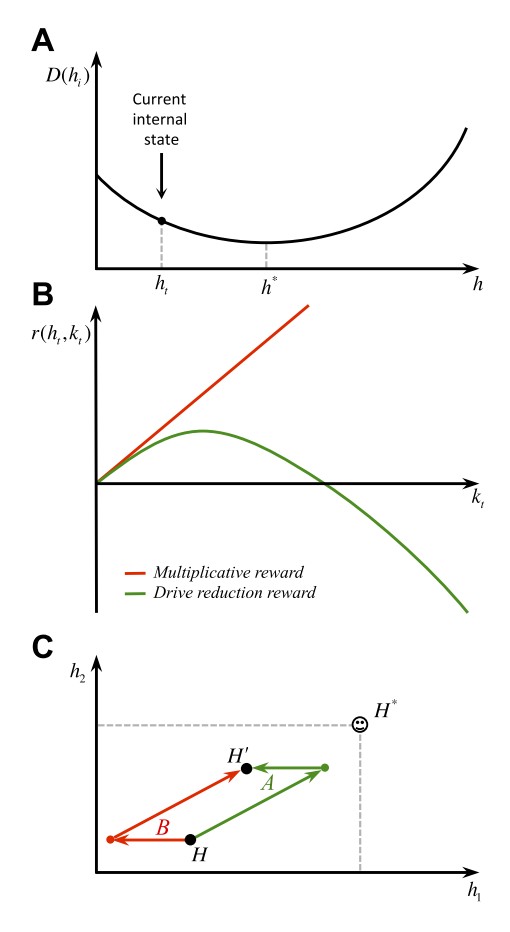

**Figure 11**. Behavioral predictions of the model. (**A**) Differential predictions of the multiplicative (linear) and drive-reduction (non-linear) forms of reward. In our model, assuming that the internal state is at $h_t$ (**A**), outcomes larger than $h^* - h_t$ result in overshooting the setpoint and thus a declining trend of the rewarding value (**B**). Previous models, however, predict the rewarding value to increase linearly as the outcome increases in magnitude. (**C**) Our model predicts that when given a choice between two options with equal net effects on the internal state, animals choose the option that first results in reducing the homeostatic deviation and then is followed by an increase in deviation (green), as compared to a reversed-order option (red).

levels of analysis (*Marr, 1982*): the free-energy theory is a computational framework, whereas our theory fits in the algorithmic/representational level. In the same line, the two theories use different mathematical tools as their optimization techniques. The free energy approach uses variational Bayes inference. Thus, rationality in that model is bounded by the simplifying assumptions for doing 'approximate' inference (namely, factorization of the variational distribution over some partition of the latent variables, Laplace approximation, etc.). Our approach, however, depends on tools from optimal control theory and thus, rationality is constrained by the capabilities and weaknesses of the variants of the RL algorithm being used (e.g. model-based vs. model-free RL). In this sense, while the notion of reward is redundant in the free energy formulation, and physiological stability is achieved through gradient descent function, homeostasis in our model can only be achieved through computing reward. In fact, the associative learning component in our model critically depends on receiving the approximated reward from the upstream regulatory component. As a result, our model remains faithful to and exploits the well-developed conditioning literature in behavioral psychology, with its strengths and weaknesses.

A further approach toward adaptive homeostatic regulation is the predictive homeostasis (otherwise known as allostasis) model (*Sterling, 2012*) where the classical negative-feedback homeostatic model is coupled with an inference system capable of anticipating forthcoming demands. In this framework, anticipated demands increase current homeostatic deviation (by adjusting the setpoint level) and thus, prepare the organism to meet the predicted need. Again, the concept of reward is redundant in this model and motivated behaviors are directly controlled by homeostatic deviation, rather than by *a priori* computed and reinforced rewarding values.

As alternative to the homeostatic regulation theories phrased around maintenance of setpoints, another theoretical approach toward modeling regulatory systems is the 'settling point' theory (*Wirtshafter & Davis, 1977*; *Berridge, 2004*; *Müller et al., 2010*; *Speakman et al., 2011*). According to this theory, by viewing organisms as dynamical systems, what looks like a homeostatic setpoint is just the stable state of the system caused by a balance of different opposing effectors on the internal variables. However, one should notice that mathematically, such dynamical systems can be re-formulated as a homeostatically regulated system, by writing down a potential functional for the system (or an energy function). Such an energy function is equivalent to our drive function whose setpoint corresponds to the settling point of the dynamical system formulation. Thus, there is equivalence between the two methods, and the setpoint approach summarizes the outcome of the underlying dynamical system on the regulated variables. Note that nothing precludes our framework to treat the setpoint conceptually as maintained internally by an

underlying system of effectors and regulators. However, the setpoint/drive-function formulation conveniently allows us to derive our normative theory.

## Predictions

Here we list the testable predictions of our theory, some of which put our model to test against alternative proposals. Firstly, as mentioned before (*Figure 9*), our theory predicts that the oral vs. fistula proportion in the water self-administration task (*McFarland, 1969*) affects the speed of satiation: the higher the oral portion is, the faster the setpoint will be reached.

Secondly, as discussed before, our model predicts an inverted U-shaped utility function (*Figure 11A,B*). This is in contrast to the multiplicative formulations of deprivation-modulated reward.

Thirdly, our model predicts that if animals are offered with two outcomes where one outcome reduces the homeostatic deviation and the other increases the deviation, the animal chooses to first take the deviation-reducing and then the deviation-increasing outcome (*Figure 11C*, green sequence), but not the other way around (*Figure 11C*, red sequence). This is due to the fact that future deviations (and rewards) are discounted. Thus, the animal tries to postpone further deviations and expedite drive-reducing outcomes.

Fourthly, as explained earlier, we predict that animals are capable of learning not only Pavlovian, but also instrumental anticipatory responding. This is in contrast to the prediction of the predictive homeostasis theory (*Woods & Ramsay, 2007*; *Sterling, 2012*; Stephen C ).

Finally, our theory predicts that upon reducing the magnitude of the outcome, a transitory burst of responding should be observed. We simulate both our model (*Figure 12*, left) and classical homeostatic regulation models (*Figure 12*, right) in an artificial environment where pressing a lever results in the agent receiving a big outcome (1 g) during the first hour, and a significantly smaller outcome (0.125 g) during the second hour of the experiment. According to the classical models, the corrective response (lever-press) is performed when the internal state drops below the setpoint. Thus, during the first hour, the agent responds with a stable rate (*Figure 12E,F*) in order maintain the internal state above the setpoint (*Figure 12D*). Upon decreasing the dose, the agent waits until the internal state again drops below the setpoint. Thereafter, the agent presses the lever with a new rate, corresponding to the new dose. Therefore, according to this class of models, response rate switches from a stable low level to a stable high level, with no burst phase in between (*Figure 12F*).

According to our model, however, when the unit dose decreases from 1 g to 0.125 g, the agent requires at least some new experiences with the outcome in order to realize that this change has happened (i.e., in order to update the expected outcome associated with every action). Thus, right after the dose is decreased, the agent still expects to receive a big outcome upon pressing the lever. Therefore, as the objective is to minimize deviation from the setpoint (rather that staying above the setpoint), the agent waits for a period equal to the normal inter-infusion interval of the 1 g unit-dose. During this period, the internal state reaches the same lower bound as in previous trials (*Figure 12A*). Afterward, when the agent presses the lever for the first time, it receives an unexpectedly small outcome, which is not sufficient for reaching the setpoint. Thus, several further responses will be needed to reach the setpoint, resulting in a burst of responding after decreasing the unit dose (*Figure 12B,C*). After the setpoint is achieved, the agent presses the lever with a lower (-than-burst) rate, in order to keep the internal state close to the setpoint. In sum, in contrast to the classical HR models, our theory predicts a temporary burst of self-administration after dose reduction (See *Figure 12—source data 1* for simulation details).

## Limitations and future directions

From an evolutionary perspective, physiological stability and thus survival may themselves be seen as means of guaranteeing reproduction. These intermediate objectives can be even violated in specific conditions and be replaced with parental sacrifice. Still, we believe that homeostatic maintenance can explain a significant proportion of motivated behaviors in animals. It is also noteworthy that our theory only applies to rewards that have a corresponding regulatory system. How to extend our theory to rewards without a corresponding homeostatic regulation system (e.g., social rewards, novelty-induced reward, etc.) remains a key challenge for the future.

In order to put forth our formal theory we had to put forward several key constraints and assumptions. As further future directions, one could relax several constraining assumptions of our formal setup of the theory. For example, redesigning the model in a *partially observable* condition (as opposed to the fully-observable setup we used) where the internal state observation is susceptible to noise could have

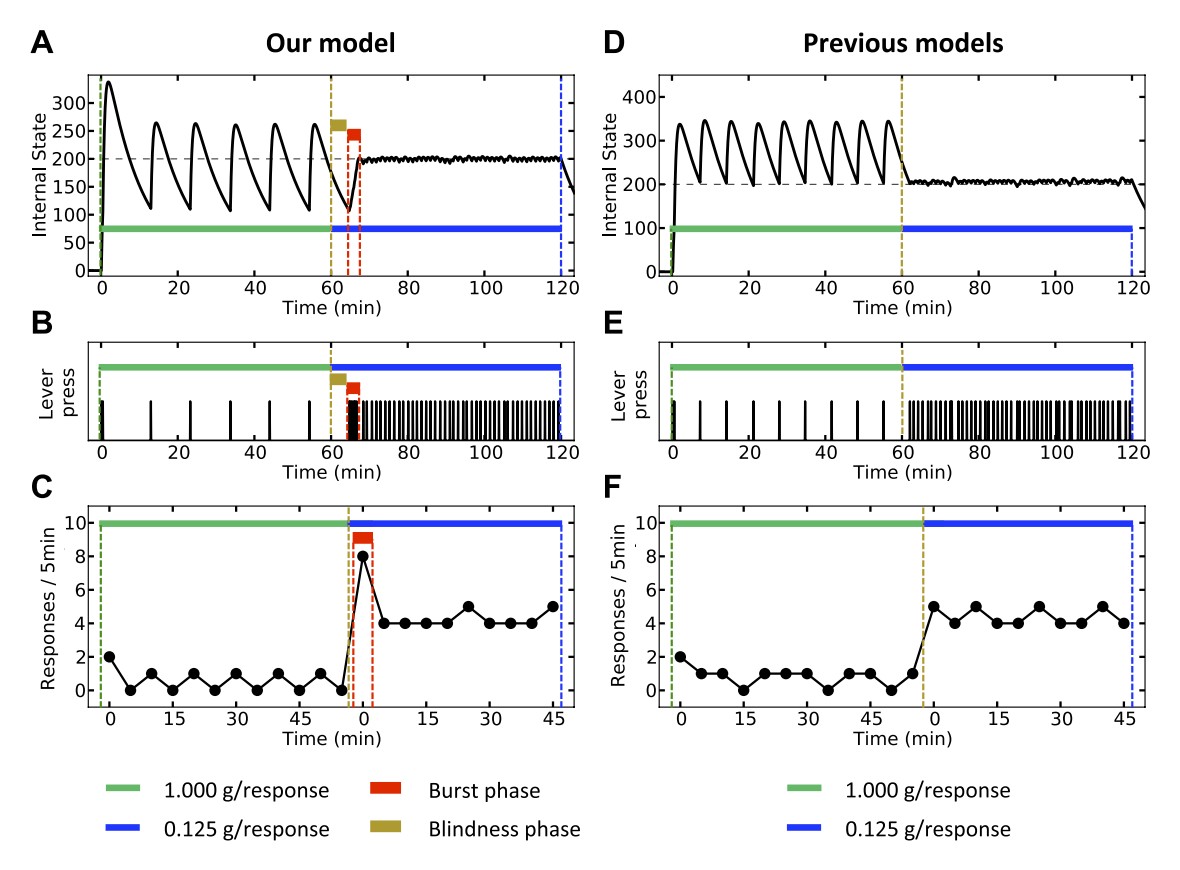

**Figure 12**. Simulation results, predicting a transitory burst of responding upon reducing the dose of outcome. Our model (left column) and negative-feedback models (right column) are simulated is a process where responding yields big and small outcomes, during the first and second hours of the experiment, respectively. In our model, the objective is to stay as close as possible to the setpoint (**A**), whereas in previous homeostatic regulation models, the objective is to stay above the setpoint (**D**). Thus, our model predicts a short-term burst of responding after the dose reduction, followed by regular and escalated response rate (**B**, **C**). Classical HR models, however, predict an immediate transition from a steady low to a steady high response rate (**E**, **F**). See *Figure 12—figure supplements 1* and *Figure 12—source data 1* for simulation details.

The following source data and figure supplements are available for figure 12:

**Source data 1**. Free parameters for the within-session dose-change simulation.

**Figure supplement 1**. The Markov Decision Process used for the within-session dose-change simulation.

important implications for understanding some psychiatric diseases and self-perception distortion disorders, such as anorexia nervosa. Also, relaxing the assumption that the setpoint is fixed and making it adaptive to the animal's experiences could explain tolerance (as elevated perception of desired setpoint) and thus, drug addiction and obesity. Furthermore, relaxing the restrictive functional form of the drive function and introducing more general forms could explain behavioral patterns that our model does not yet account for, like asymmetric risk-aversion toward gains vs. losses (*Kahneman & Tversky, 1979*).

## Conclusion

In a nutshell, our theory incorporates a formal physiological definition of primary rewards into a novel homeostatically regulated reinforcement learning theory, allowing us to prove that economically rational behaviors ensure physiological integrity. Being inspired by the classic drive-reduction theory of motivation, our mathematical treatment allows for quantitative results to be obtained, predictions that make the theory testable, and logical coherence. The theory, with its set of formal assumptions and proofs, does not purport to explain the full gamut of animal behavior, yet we believe it to be a credible step toward developing a coherent mathematical framework to understand behaviors that

depend on motivations stemming from internal states and needs of the individual. Furthermore, this work puts forth a meta-hypothesis that a number of apparently irrational behaviors regain their rationality if the internal state of the individual is taken into account. Among others, the relationship between our learning-based theory and evolutionary processes that shape animal a priori preferences and influence behavioral patterns remains a key challenge.

# Materials and methods

## Rationality of the theory

Here we show analytically that maximizing rewards and minimizing deviations from the setpoint are equivalent objective functions.

### Definition:

A 'homeostatic trajectory', denoted by $p = \{K_0, K_1, K_2, \ldots\}$, is an ordered sequence of transitions in the $v$-dimensional homeostatic space. Each $K_i$ is a $v$-dimensional vector, determining the length and direction of one transition. We also define $\mathcal{P}(H_0)$ as the set of all trajectories that if start from $H_0$, will end up at $H^*$.

### Definition:

For each homeostatic trajectory $p$ that starts from the initial motivational state $H_0$ and consists of $w$ elements, we define $SDD_p(H_0)$ as the 'sum of discounted drives' through that trajectory:

$$SDD_p\left(H_0\right) = \sum_{t=0}^{w-1} \gamma^t . D(H_{t+1}) \tag{S1}$$

Where $\gamma$ is the discount factor, and $D(.)$ is the drive function. Also, starting from $H_0$, the internal state evolves by $H_{t+1} = H_t + K_t$.

### Definition:

Similarly, for each homeostatic trajectory $p$ that starts from the initial motivational state $H_0$ and consists of $m$ elements, we define $SDR_p(H_0)$ as the 'sum of discounted rewards' through that trajectory:

$$SDR_p\left(H_0\right) = \sum_{t=0}^{w-1} \gamma^t . r_t = \sum_{t=0}^{w-1} \gamma^t . \left(D\left(H_t\right) - D\left(H_{t+1}\right)\right) \tag{S2}$$

### Proposition:

For any initial state $H_0$, if $\gamma < 1$, we will have:

$$\underset{p \in \mathcal{P}(H_0)}{\operatorname{argmin}} \; SDD_p\left(H_0\right) = \underset{p \in \mathcal{P}(H_0)}{\operatorname{argmax}} \; SDR_p\left(H_0\right) \tag{S3}$$

Roughly, this means that a policy that minimizes deviation from the setpoint, also maximizes acquisition of reward, and vice versa.

### Proof:

Assume that $p_i \in \mathcal{P}(H_0)$ is a sample trajectory consisting of $w_i$ transitions. As a result of these transitions, the internal state will take a sequence like: $\{H_{i,0} = H_0, H_{i,1}, H_{i,2}, \ldots, H_{i,w} = H^*\}$. Denoting $D(H_x)$ by $D_x$ for the sake of simplicity in notation, the drive value will take the following sequence: $\{D_{i,0} = D_0, D_{i,1}, D_{i,2}, \ldots, D_{i,w} = D^* = 0\}$. We have:

$$SDD_{p_i}\left(H_0\right) = D_{i,1} + \gamma . D_{i,2} + \gamma^2 . D_{i,3} + \ldots + \gamma^{w-1} . D^* \tag{S4}$$

We also have:

$$
\begin{aligned}
SDR_{p_i}\left(H_0\right) &= r_{i,0} + \gamma . r_{i,1} + \gamma^2 . r_{i,2} + \ldots + \gamma^{w-1} . r_{i,w-1} \\
&= \left(D_0 - D_{i,1}\right) + \gamma . \left(D_{i,1} - D_{i,2}\right) + \gamma^2 . \left(D_{i,2} - D_{i,3}\right) + \ldots + \gamma^{w-1} . \left(D_{i,w-1} - D^*\right) \\
&= D_0 + (\gamma - 1) . \left(D_{i,1} + \gamma . D_{i,2} + \gamma^2 . D_{i,3} + \ldots + \gamma^{w-2} . D_{i,w-1}\right) \\
&= D_0 + (\gamma - 1) . \; SDD_{p_i}\left(H_0\right)
\end{aligned}
\tag{S5}
$$

Since $D_0$ has a fixed value and $\gamma - 1 < 0$, it can be concluded that if a certain trajectory from $\mathcal{P}(H_0)$ maximizes $SDR(H_0)$, it will also minimize $SDD(H_0)$, and vice versa. Thus, the trajectories that satisfy these two objectives are identical.

## Hyper-palatability effect

For the especial case that m/n = 1, *Equation 11* can be rewritten as follows:

$$
\begin{aligned}
r(H_t, K_t) &= D(H_t) - D(H_t + K_t) + T \\
&= (H_t - H^*)^2 - (H_t + K_t - H^*)^2 + T \\
&= \left(H_t - \left(H^* + \frac{T}{2K_t}\right)\right)^2 - \left(H_t + K_t - \left(H^* + \frac{T}{2K_t}\right)\right)^2
\end{aligned}
$$

(S6)

This means that the effect of $T$ is equivalent to having a simple HRL system (without term $T$) whose drive function is shifted such that the new setpoint is equal to $H^* + \frac{T}{2K_t}$, where $H^*$ is the setpoint of the original system. This predicts that the bigger the hyper-palatability factor $T$ is, the higher the new steady state is, and the higher the real nutritional content $K_t$ of the food outcome is, the less divergence of the new setpoint from the original setpoint is.

*Equation 5* can also be re-written as:

$$
\begin{aligned}
r(H_t, K_t) &= D(H_t) - D(H_t + K_t) + T \\
&= (H_t - H^*)^2 - (H_t + K_t - H^*)^2 + T \\
&= \left(\left(H_t - \frac{T}{2K_t}\right) - H^*\right)^2 - \left(\left(H_t - \frac{T}{2K_t} + K_t\right) - H^*\right)^2
\end{aligned}
$$

(S7)

This can be interpreted as the effect of $T$ being equivalent to a simple HRL system (without term $T$) whose internal state $H_t$ is underestimated by $\frac{T}{2K_t}$ units. That is, hyper-palatability makes the behavior look like as if the subject is hungrier than what they really are.

## Acknowledgments

We thank Peter Dayan, Amir Dezfouli, Serge Ahmed, and Mathias Pessiglione for critical discussions, and Peter Dayan and Oliver Hulme for commenting on the manuscript. The authors acknowledge partial funding from ANR-10-LABX-0087 IEC (BSG), ANR-10-IDEX-0001-02 PSL* (BSG), CNRS (BSG), INSERM (BSG), and FRM (MK). Support from the Basic Research Program of the National Research University Higher School of Economics is gratefully acknowledged by BSG.

## Additional information

### Funding

| Funder | Grant reference number | Author |
| --- | --- | --- |
| Gatsby Charitable Foundation | | Mehdi Keramati |
| National Research University Higher School of Economics | Basic Research Program | Boris Gutkin |
| Institut national de la santé et de la recherche médicale | INSERM U960, France | Boris Gutkin |
| Center for Research and Interdisciplinary | Frontiers du Vivant | Mehdi Keramati |
| Agence Nationale de la Recherche | ANR-10-LABX-0087 IEC, France | Boris Gutkin |

| Funder | Grant reference number | Author |
|---|---|---|
| Agence Nationale de la Recherche | ANR-10-IDEX-0001-02 PSL, France | Boris Gutkin |

The funders had no role in study design, data collection and interpretation, or the decision to submit the work for publication.

### Author contributions

MK, Doing simulations, Deriving analytical proofs, Conception and design, Analysis and interpretation of data, Drafting or revising the article; BG, Discussing the results, Drafting or revising the article

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
