## [Decision Letter]

[Editors’ note: this article was originally rejected after discussions between the reviewers, but the authors were invited to resubmit after an appeal against the decision.]

Thank you for choosing to send your work entitled “Collecting reward to defend homeostasis: A homeostatic reinforcement learning theory” for consideration at eLife. Your full submission has been evaluated by a Senior editor and 3 peer reviewers, and the decision was reached after discussions between the reviewers. We regret to inform you that your work will not be considered further for publication at this stage.

While the topic of your manuscript is potentially very interesting, the reviewers and BREs consulted and had a number of substantive issues that enter into this decision.

During our initial BRE discussion, one pointed out that humans (and many animals) indulge in many behaviors that are not in the animal's best interest, and violate the premises of physiological homeostasis. For example, obesity and drug-taking behavior, and many of our other activities are clearly not physiologically homeostatic. This appears to be an issue that calls into question some of the fundamental assumptions of this work? If not, how does this come into play?

*eLife* does not allow supplemental material as is common by many journals. Instead, all material of substance should be incorporated into the main text, and less important material omitted. eLife has no specific length limitations, and this policy is to support papers that present an integrated story.

We are including the reviews in entirety below for your information. eLife welcomes theoretical work when it can add new insight into interesting biological problems, which is why we chose to review it. We only recommend revision when there is a straightforward path that we foresee could lead to a successful outcome, which is not obvious in the case of this manuscript. Consequently, we are returning it to you so that you can submit it elsewhere, either as is, or benefiting from this review. Because the reviewers find this work potentially interesting, if you feel you can successfully craft a new manuscript that addresses the issues raised in this review, we would be willing to consider it as an entirely new submission, which would be reviewed either by the same or different reviewers, and which would not be guaranteed to be successful.

*Reviewer #1*:

I enjoyed reading this interesting discussion reinforcement learning in the setting of homeostasis. I thought your treatment was both formal and scholarly. It usefully highlights the fact that reinforcement learning or optimal control can be applied to homeostatic regulation. Having said this, as the author of the free energy principle, I find the notion that optimal control (e.g. Dynamic programming or reinforcement learning) can be applied to physiological homeostasis a little self-evident. I think the deeper challenge is to provide a principled explanation for why optimal control emerges from, or is mandated by, homeostasis.

If I understand your idea correctly, you are saying that applying optimal control to the deviation of homeostatic variables from their set point accounts for some key empirical findings in behavioural psychology. I think this is a perfectly fine thing to say and a useful contribution. However, your focus on reinforcement learning is a bit colloquial. One could equally suggest that applying predictive coding to homeostatic deviations (prediction errors about their set point) is a plausible explanation for the empirical findings. This is because one can formulate optimal control as a Kalman filter (exactly for linear quadratic control) and predictive coding is a biologically plausible implementation of Bayesian filtering.

This is important because unlike reinforcement learning, predictive coding provides a process theory or mechanistic explanation for how the brain works. In other words, it is not just a normative model but makes mechanistic predictions. I stress predictive coding because there is a lot of work on homeostasis and interoceptive inference based upon predictive coding (within the larger setting of minimising variational free energy). For example, Anil Seth has a several articles that you might want to refer to. I think you also need to discuss the intellectual background to your work in cybernetics (and more recently synergetics). A key example here would be the good regulator theorem stemming from the work of Ross Ashby on self organisation and his notion of a homeostat. A more recent (and possibly heuristic) formulation of these ideas can be found in the literature on allostasis. I notice that you refer to allostasis in the last sentence of your paper (and also cite Peter Sterling). However, you never define the distinction between allostasis and homeostasis and how this relates to the anticipatory aspects of your formulation (e.g., temporal discounting within control theory).

The other issue I think you need to discuss and qualify is the status of your normative model. Temporal discounting coefficients and other arbitrary parameters (for example the n and m in [Disp-formula equ1]) are characteristic of reinforcement learning models, which undermines their normative status. I am assuming here that normative means that one can describe a process in terms of optimising an objective function. However, adding ad hoc parameters to the function destroys any uniqueness or optimality properties of the normative explanation. An example of this is the temporal discounting factor that your treatment emphasises. In normative Bayesian accounts, this reflects the precision of random fluctuations in hierarchical or volatility models of contingencies. Crucially, for any given outcomes, there is a Bayes-optimal temporal discounting that renders the Bayesian description truly normative.

My main point here is that I think you need to discuss the broader church of theoretical approaches to homeostasis and self organisation and highlight why reinforcement learning might provide a useful focus. I would not be shy about emphasising its shortcomings and pointing to outstanding conceptual challenges. In one sense, making reinforcement learning accountable to homeostatic imperatives is one step in this direction, as illustrated by the importance of temporal discounting and converting a homeostasis into allostasis (and highlighting the fact that reinforcement learning is not context sensitive). As a minimum, I think you should discuss the good regulator theorem and early cybernetic formulations. I think you should also mention the notion of interoceptive inference or prediction as a relevant example of more generic Bayesian approaches. You may find useful references in the following:

Seth AK. Interoceptive inference, emotion, and the embodied self. Trends Cogn Sci. 2013 Nov;17(11):565-73.

http://en.wikipedia.org/wiki/The_free_energy_principle

http://en.wikipedia.org/wiki/Allostasis

http://en.wikipedia.org/wiki/Homeostat

1) You introduce reinforcement learning. It might be useful to comment upon its biological plausibility. Are there any detailed models of how neuronal circuits would implement reinforcement learning in the context of homoeostasis?

2) In [Disp-formula equ1], you should highlight the fact that n and m are free parameters It would be useful to say something like:

Note that m and n (both greater than 2) are free parameters that have an important nonlinear effect on the mapping between homeostatic deviations and their motivational consequences. Later, we will consider this mapping in terms of risk and classical utility theory (risk aversion).

3) I thought that your argument was confused (or written in a confusing way). You seem to imply that in the absence of discounting, the value of a policy depends only on the initial and final states regardless of its trajectory. This is not the case in standard applications of optimal control or reinforcement learning. The quantity that is optimised is the expected reward at each point (in free energy formulations this would be the path integral). This means that the path becomes important in determining the expected reward or value. Furthermore, there is no final state unless you are considering finite horizon problems. Perhaps you meant there is no temporal integration into the future at all?

*Reviewer #2*:

The authors describe a model in which reward is operationally defined and computed according to the degree to which an outcome reduces the distance between the animal's current state and an idealized homeostatic state (setpoint). The authors characterize this as a drive-reduction model and revisit several points of controversy from the 1940s/50s surrounding Hull's drive-reduction theory. The authors seem to suggest that (a) their model addresses all the criticism of drive-reduction and (b) that more contemporary views of motivation, which have largely rejected Hull's model, have been hasty and that we should reconsider the drive-reduction model-rescue Hull, as the authors put it.

Kudos to the authors for tackling an interesting and really important problem: how is a stimulus evaluated to determine whether it is rewarding (or aversive) in the first place, an issue almost entirely ignored in contemporary reinforcement learning models. To address the question of how are stimuli evaluated and to place that into a homeostatic framework in which reward is contingent upon the state of the animal is an important endeavor and I was excited about the authors' model.

However, several issues severely temper my enthusiasm, as follows:

1) Framing the entire model, not just the Introduction and Discussion but the Results, around revisiting and apparently rekindling a half-century old debate between drive and incentive theories of motivation seems unfortunate. The result is that the manuscript reads more like a review, mostly comprised of arguments that border on polemical. This leads to several problems:

(a) The Results section does not read like a Results section but like a Discussion section. For each of the subsections in the Results I would expect the question/problem to be defined clearly, then the details of how this was addressed in a simulation and then the results of the simulation (for example, one expects results to be substantively organized around/referencing figures). The 'results', as the manuscript stands, are essentially presented in brief in the figure legends. Less argument and 'defending' the model and more illustrating its function.

(b) Instead of framing this in a half century old debate, why not anchor it in contemporary scientific issues and questions? How are TD models being developed and applied, what are their short-comings in particular applications, how does this model improve upon that state of the art?

(c) The manuscript ends up with a cavalier quality of 'our model has just solved all these problems' but the solution is superficial (see below).

(d) The manuscript is too ambitious, attempting to implement a simple computational model of drive-reduction and then, in effect, wanting to 'put to rest' what comprised decades of (much still unresolved) controversy. Again, the result reads more like a polemical review, both cavalier and superficial. A more focused, more empirical, less argumentative approach might be more successful.

2) Several arguments are not convincing. Example; extinction burst. The intuitive explanation for extinction bursts is that the animal expects a reward and when it does not arrive, continues to work/press because prior learning does not unravel (or update) that quickly, especially if reward was stochastic in the first place (and, well, the animal is still hungry). The suggestion that a mouse that does not get a reward after pressing the lever 10 times would, as a result, be in a physiological state significantly further from setpoint that this increases their responding seems unlikely. If the mouse presses the lever 10x in 2 minutes (slow for a well-trained mouse), then the mouse's hunger state would only be increased by 2 min metabolism, but mouse behavior is not that regimented nor does receiving a pellet significantly reduce pressing when not in extinction (would you expect to see increased responding if you put the mouse in 2 min later today than yesterday?)

3) Confusion on drive-reduction vs. incentive. In the anticipatory responding section, the authors argue that their model explains why an animal would respond in the absence of a deficit state: the animal has previously learned that a cue or action is rewarding, i.e., that it is associated with relieving a deficit state. Thus, the authors argue, that learning induces the animal to respond even when not in a deficit state. But this is, in fact, precisely what incentive salience is: that animals learn the value of stimuli and that this learned value induces responding in the absence of an actual deficit state. And, in fact when animals respond for such learned value, they are not motivated but a current deficit state (as drive-reduction would have) but by learned incentive, which is causing the behavior in the absence of such need. The authors have merely reinvented the incentive-salience wheel and don't appear to realize it.

4) Anatomical substrates. There is much data suggesting that metabolic state and nutritional information is signaled in the brain (how could it not be?) and, moreover, that this impacts the dopamine system. However, the authors make a jump here and assume, without evidence, that this is a homeostatic system. Because metabolic information is signaled does not necessitate homeostasis. Equally important, as the authors are discussing the effect of metabolic state on dopamine signaling, a distinction between tonic and phasic needs to be made. So for example, tonic dopamine goes up and remains up for a period of time during a meal, how does this fit in with what should be decreased signaling as the state of the system moves toward the setpoint (i.e., with each bite)? In extinction bursts, the authors seem to suggest that this sort of micro, moment to moment variation in state is signaled. Is there evidence to support that hunger will change the magnitude of the reward?

Requested brevity of review limits detailed critique, but each and every section essentially presents the same sorts of problems (do not make a compelling case, don't actually solve the problem). More generally, what have the authors really done? They have added into a TD model a computational mechanism for determining what is and is not rewarding. That, I think, could represent a substantial contribution if the authors could focus specifically on that and develop it in the context of contemporary models/problems. When they attempt to resolve one of the largest, most complex controversies in all of the history of psychology in one short manuscript, they fall into trouble. From the point of view of computational modeling, is this not simply a form of utility? How is it different from other models that have incorporated utility? This seems the more relevant context for developing and presenting the model. As for drive-reduction and rescuing Hull, surely Hull and his followers meant 'ideal_state MINUS current_state = motivation'? Do the authors really believe that if only Hull had had RL learning theory in which to embed his drive-reduction, the history of psychology would have been different? And now it can be corrected?

I think the effort and intent is to be commended: how reward is evaluated and how learning and motivation is state-dependent is a critical question not well developed in RL models. And starting with a simple drive-reduction model is a reasonable starting point, but I would suggest that rather than implementing a drive reduction model and saying 'viola, a half century of controversy resolved', the authors might make a far better contribution by asking what problems are solved and what problems remain or are created; that is, to see this model as a starting point of something really important, not an end point.

*Reviewer #3*:

The authors implement, in a simple mathematical manner, an elaboration of Hull's “drive reduction theory” (1943), a theory that reinforces a behavior if it reduces any deviation of an organism from homeostasis. As far as I can tell, the specific elaboration is to incorporate more modern theories of reinforcement learning as the mechanism to reduce the “drive” and in so doing manage to alleviate a couple of criticisms of Hull's work (e.g., how secondary reinforcers such as money can operate) as well as account for a number of behavioral phenomena. The impressive aspect of this paper is that it provides a unifying theory of motivation that explains several behavioral phenomena; some of them the authors claim have no other explanation to date.

Unfortunately I find the paper rather abstract, with the concrete details that are presented being a “convenient fiction” (such as “distance of the internal state from the set point” in a fictional “homeostatic space”) that do not directly correspond to any biological property. I focus on one particular result by way of example:

The authors claim that “temporal discounting” (the reduced value of a behavior as its benefit is less immediate) is explained within their theory (for the first time) as “in order to maintain internal stability, it is necessary to discount future rewards.” However, the derivation underlying this argument seems to rely on the assumption that all paths in homeostatic space are possible – e.g., one could raise one's body temperature to 45C then return it back to 37C, so in order to avoid such a path the positive value of returning from 45C to 37C should be discounted compared to the preceding negative value of rising from 37C to 45C. In my mind such an argument seems bizarre, since once reaching 45C the organism can never return to its “sweet spot”; mathematically the “homeostasis space” defined by the authors is not really “curl-free” or without bifurcations, because routes matter, homeostatic fixed points change etc. So the solution seems to explain a problem that only arises in the proposed formalism. Alternative factors – such as the environment being less and less predictable as one moves to the future and that uncertainties in the consequences of behaviors increase as one projects into the future – seem a much more plausible explanation of temporal discounting.

In general, I question whether the theory is falsifiable as I do not see specific testable predictions and the quantitative results presented appear to be produced purely by fine tuning of unconstrained parameters. Also, the authors make claims such as “the major result of our theory, which is that the rationality of behavioral patterns is geared toward maintaining physiological stability”, but that was Hull's theory and is hardly novel or a surprising idea. Since the authors add a mathematical framework to prior theories and their combination, the paper would greatly benefit (and in my view could be acceptable) if it could make some concrete quantitative predictions of the results of behavioral experiments that could be tested, or at a minimum, when numbers in their model are fit to one data set, they show they can reproduce other data without extra fitting.

[Editors' note: further revisions were requested prior to acceptance, as described below.]

Thank you for sending your work entitled “Collecting reward to defend homeostasis: A homeostatic reinforcement learning theory” for consideration at eLife. Your article has been favorably evaluated by Eve Marder (Senior editor) and 3 reviewers.

As you will see below, all of the reviewers find this version vastly improved over your previous submission, and all of the reviewers are quite positive about this work at this point. Nonetheless, each of them has some specific suggestions for editorial revision, mostly to do with emphasis and presentation. Because these are well-articulated in the actual reviews, I am taking the unusual tactic of enclosing these reviews in their entirety, as they were meant constructively. I hope that you will find them helpful in making a considerably improved piece of work more transparent and accessible.

*Reviewer #1*:

The authors have embarked on the valuable task of producing a computational framework that combines theories of reinforcement learning with those of homeostasis and drive reduction. This is a worthwhile goal and the authors have several examples of behaviors that arise within their framework as well as predictions. I do think the manuscript reads a bit as though come of the ideas of combining reinforcement learning and homeostasis are novel to the authors, whereas in reality their contribution is to add a mathematical/computational framework which allows for quantitative predictions to be made and suggests what could/should be observed in any neural mechanism.

While overall the writing is very clear, I think the manuscript would be served by the authors being more careful to tone down statements that suggest the idea of combining homeostasis and reinforcement learning is their own. After all, everyone knows that when one is out on a cold winter's day a hot drink is rewarding, whereas in the middle of a hot summer's day a cold drink has greater rewarding value. The authors deserve credit for developing a mathematical scheme (the first I think?) where such results fall out, and I think they now have enough quantitative results and predictions that make the scheme testable.

In a similar vein, some statements to motivate the work are exaggerated, for example in the first line of Discussion the authors’ state:

“Theories of conditioning are founded on the argument that animals seek reward, while reward is defined as what animals seek.”

I think that while these definitions can be found, to state simply “reward is defined” without adding “by some” or “can be defined” or “has been defined by some” is too bold and general. One can find plenty of definitions of reward, in which “primary reward” is “that which aids survival” or “helps propagate the species” or simply in general English, reward is something that is good for you!

In a couple of places (including the Abstract) the authors state that they:

“prove analytically that reward-seeking and physiological stability are two sides of the same coin” and “Our theory mathematically proves that seeking rewards is equivalent to the fundamental objective of physiological stability“ whereas in fact through their definition of drive;”we define the “drive” as the distance of the internal state from the setpoint“ the authors assume this to be the case and develop a mathematical theory where this result is true. One must be careful in mathematical proofs as to what are the premises. Since the rewards associated with sexual desire are outside the model (as the authors comment) it is clear that it is only within their theory that the mathematical “proof” holds.

*Reviewer #2*:

The authors have dramatically transformed this paper from the last submission. I like it a lot and think it has some important things to say. It feels much more anchored in the modern RL literature and the discussion of Hull is much more nuanced and realistic. However, there are still some comments:

1) The authors make a comment early on that equates reward/reinforcer/utility. Given the obvious sophistication of the authors, this is unfortunate. In particular, to make clear the relationship between prior treatments of utility and the authors’ proposal would be helpful. Notably, the authors do describe other approaches to this, but even a sentence or two early on that clarifies rather than lumps together the difference between reinforcer/utility. Specifically because the authors are essentially arguing that homeostatic utility determines reinforcement properties.

2) The authors make a comment about 'erroneous estimation of error' and later in the manuscript talk at length about, essentially, taste serving as cues. Three lines of investigation that the authors might find useful in this discussion: (1) Beeler et al Eur J Neuroscience 2012 'taste uncoupled from nutrition fails to sustain the rewarding properties of . . . ' (2) the work of Swithers with artificial sweeteners:

Swithers, S.E. & Davidson, T.L. (2008) A role for sweet taste: calorie predictive relations in energy regulation by rats. Behav. Neurosci., 122, 161- 173.

Swithers, S.E., Baker, C.R. & Davidson, T.L. (2009) General and persistent effects of high-intensity sweeteners on body weight gain and caloric compensation in rats. Behav. Neurosci., 123, 772-780.

Swithers, S.E., Martin, A.A. & Davidson, T.L. (2010) High-intensity sweeteners and energy balance. Physiol. Behav., 100, 55-62.

Finally, the authors cite one paper by de Araujo, but he has significantly developed the notion that the DA cells specifically serve as a metabolic sensor.

Other than that, I think there are many things that one could nitpick about, especially with regards to the endless details and nuances of the model (eg., I am not sure the authors have fully addressed the question the other reviewer had regarding the 'shortest distance between two points' idea). However, I think the paper is interesting, brings up some very good points, is well done and, as the authors point out, targets the mutual weakness of HR and RL models and brings them together nicely.

*Reviewer 3*:

This is an improved version of a previous submission. I see merit in the ideas behind this work. However, I think the authors still could communicate their thoughts in a more structured way, and have made some suggestions below.

This is a much improved version of a previous submission to eLife. It basically connects homeostatic imperatives with classical (utilitarian and control theoretic) formulations of adaptive behaviour. There is a central technical result that links homeostasis to discounted future reward, which the authors exploit to explain a number of phenomena in the reinforcement learning literature. The authors have contextualised their contribution in relation to other (theoretical) frameworks. There are some outstanding issues with the way that the authors structure their paper.

Major points:

1) Scientifically, I think you need to highlight and unpack the major result in the appendix. At an appropriate point in the main text, I would include a paragraph of the following sort:

“In summary, we have established a formal link between the homeostatic imperatives to keep physiological states near some set point and the maximisation of temporally discounted reward (or minimisation of some loss function). This is an important and non-trivial result. The appendix provides a formal proof; however, the underlying idea is fairly simple. Imagine you had to plan a hill walk, during which you wanted to maximise the height (altitude or reward) averaged over the path you take. If someone dropped you at the bottom of the hill, the optimum path would be to ascend the hill and spend as long as possible at the top before returning to your pick up point. Notice that this entails ascending the hill (reward function) before descending. Implicit in this strategy is a maximisation of temporally discounted reward. In other words, going up the hill first and then coming down is better than going down and then coming back up. It is this fundamental (variational) phenomenon that connects homeostasis with classical temporal discounting.

Furthermore, as indicated above, if the homeostatic cost (negative reward) is cast as a log probability then it can be treated as (free) energy. Crucially, the time average or path integral of energy is called action. This means that both the homeostasis and temporally discounted reward are ways of prescribing a principle of least action. From this perspective, one can regard the adaptive behaviours that we are trying to link as necessary and emergent properties of all dynamical systems that comply with (Hamilton's) principle of least action. We will return to this perspective in the Discussion.”

2) The second major point is about the format of your paper. It is still unclear where the reader can find the details of your simulations. I also note that you have included supplementary figures. Can I suggest that you remove all supplementary material and place it in the main text (or discard it and refer to it as results not shown). I think you should prepare the reader for the slightly unusual scientific presentation with a paragraph at the beginning of the paper along the following lines:

”We will develop our theoretical results by appealing to simulations. These simulations are described in figures (and accompanying tables) and are called upon when necessary. All the simulations in this paper followed the same procedure: first we define a model that captures the problem of interest in terms of a Markov decision process. The ensuing behaviour is then optimised using classical reinforcement learning procedures (Q-learning) to define a value function. Actions are then selected using a softmax function of the value of allowable actions or choices. For each simulation we present the graphical model or Markov decision process in the figures, along with the ensuing behaviour. Each figure is accompanied by a table specifying the parameters of the Markovian process, the Q-learning and softmax functions used to simulate behaviour.”

Note that I am suggesting, for every simulation you present, a figure and table. Whenever you refer to results that are not presented in this format I would say so explicitly so the reader does not have to wonder whether they have missed something.

3) You might want to refer to the notion of beliefs or probability distributions over excursions. In other words, the risk sensitive behaviour can also be interpreted in terms of the probability of extreme events that render the beliefs sub Gaussian; assuming the homeostatic deviation is interpreted as a log probability.

---

## [Author Response]

*Animals indulge in many behaviors that violate the premises of physiological homeostasis, like obesity and drug-taking behavior. This appears to be an issue that calls into question some of the fundamental assumptions of this work*.

We thank the BREs for prompting us to address this issue. In fact we already performed the simulations showing how irrational behaviors might arise within our theory, yet did not include it in the previous version of the manuscript as we previously felt that a full-blown treatment of irrational behaviors is beyond the scope of this paper and would merit a further publication. To address the BREs’ concern, in the present manuscript we added a subsection titled “Irrational behavior: the case of over-eating.” to illustrate (with simulations) one of the points of vulnerability of our theory that can induce irrational choices. Moreover, in the subsection “Limitations and future works” we discuss on how to approach other pathologies including drug-addiction and anorexia, as results of other mechanisms of our framework going awry. Also, as communicated to the editor previously, modeling drug addiction within our “Homeostatic Reinforcement Learning” framework has been the topic of another entire line of research in our group and we are preparing a further publication on that.

Reviewer 1:

*The relation of the theory to the free-energy framework, as well as the allostasis and the good regulator theorem to be explained. Also, the advantage of using optimal control as the optimization techniques to be discussed*.

We agree with the reviewer that our theory has significant connections with the free-energy framework. We added a subsection titled “Previous theoretical models” in order to discuss all these and other issues in detail.

*Limitations of the theory to be discussed*.

We added a subsection titled “Limitations and future works”, and detailed several limitations of the model, as well as constraining assumptions that could be eventually relaxed.

Reviewer 2:

*Framing the entire model around rekindling the Hull’s theory is unfortunate. Instead of framing this in a half-century-old debate, why not anchor it in contemporary scientific issues and questions? The results section does not read like a results section but like a discussion section*.

We thank the reviewer for this suggestion. While we stand behind our theory’s explanatory power, we do agree that the present manuscript is only a first step in addressing the modern literature on the links between motivation and the internal physiological states. Following the reviewer’s advice, we re-structured the manuscript to make it less polemic and more result-oriented, and tried to clearly de-limit its scope. Rather than framing the whole manuscript around the Hull’s theory, we now only discuss that theory and its relevancy to our model in a subsection of the Discussion titled “Relationship to classical drive-reduction theory”. Also, in that section, we withdrew the claim of “rescuing Hall”, and instead discuss the differences between our formal elaboration and the original theory (Namely, orosensory-based approximation of drive-reduction, and integration with an RL module). Also, for the sake of caution, we only claim that our model addresses “a number of significant criticisms”, rather than “all”, criticisms against the Hull’s theory.

*The Results section does not read like a Results section but like a Discussion section. For each of the subsections in the Results I would expect the question/problem to be defined clearly, then the details of how this was addressed in a simulation and then the results of the simulation (for example, one expects results to be substantively organized around/referencing figures). The 'results', as the manuscript stands, are essentially presented in brief in the figure legends. Less argument and 'defending' the model and more illustrating its function*.

In response to this concern, we transferred all the other discussion-like issues to the Discussion section (i.e., Neural substrate, predictions, previous models) and greatly expanded the Results section. In the rest of the manuscript, we only focus on the behavioral/neurobiological pattern being addressed, and mechanisms by which our model explains them. We tried to explain the replicated experiments, our simulation results, and the mechanisms of the model with much more details and clarity. Furthermore, we included the full proof of the rationality and the normativity of temporal discounting in the Methods section. In fact we considered including the full proof in the main body of the paper but felt that the paper might become too cumbersome. This is a point we are ready to discuss and if eLife might allow for a “mathbox” to be embedded within the paper, which could be a good solution.

*The manuscript ends up with a cavalier quality of 'our model has just solved all these problems' but the solution is superficial. The manuscript is too ambitious, attempting to implement a simple computational model of drive-reduction and then, in effect, wanting to 'put to rest' what comprised decades of (much still unresolved) controversy*.

*Again, the result reads more like a polemical review, both cavalier and superficial. A more focused, more empirical, less argumentative approach might be more successful*.

In general, following the points raised by the reviewer we rewrote the paper in a more empirical and less argumentative way. We further pointed out in the manuscript, which issues are resolved by our model and what are the limitations (new sections added). Furthermore, we tried to clarify in the manuscript that our modeling framework is not only a simple model of drive-reduction, but gives a normative computational theory for the interplay between the former and reinforcement learning theories of motivation. Indeed the present manuscript is only a starting point for a further development of the theory, and we tried to point this out in the manuscript. We sincerely hope that the cavalier quality of our paper has been rectified.

*The explanation for extinction bursts is not convincing*.

We see the point raised by the reviewer and understand that our suggestion is way too speculative. We decided to remove the “Extinction burst” section from the manuscript and re-consider this issue in the future.

*With respect to anticipatory responding, the proposed model is equivalent to the “incentive salience” theory*.

We thank the reviewer for pointing out this apparent lack of clarity in our paper. In the new manuscript, we discuss the differences between the two models in detail, in the subsection “Relationship to other theoretical models”.

We would also like to briefly address the specific comment of the reviewer. In our theory the value of a response depends on the internal state at the time of learning and is built from a reward definition that is based on the ability of the response to produces a drive-reducing outcome. We give a precise mathematical formulation of how this should be done (normative reinforcement learning framework). And indeed the response after learning is driven by the value as opposed to by the direct drive reduction. Previously it was controversially argued that “value” in the RL algorithms is equivalent to motivational “incentive salience” (1). However, as we could best understand, the recent computation model of incentive salience separates value learning from influences of the internal state. The value is learned as in the standard RL algorithms (with respect to a reference state and based on externally defined rewards). The internal state at the time of the response then modifies the learned value. As we now argue in the manuscript such a formulation differs from our framework and is unable to account for anticipatory responses.

*Just because metabolic information is signaled into the hypothalamus does not necessitate that it is a homeostatic system*.

We thank the reviewer for pointing out this issue. Although discussing the neural evidence for the hypothalamus being a homeostatic system is a topic of merit, we feel that in this manuscript, we would better limit the discussion to the neural evidence that is relevant to the novel contributions of our model (i.e. the “integration” of homeostatic and learning systems). We felt that a full discussion of the substrates of the two individual systems is beyond the scope of our manuscript. However, in the new manuscript, we cited further recent review articles that point to the role of hypothalamus in homeostatic regulation.

Furthermore, in the new manuscript, we have explained that from a mathematical point of view, any regulatory system can be formulated either as a dynamical system (interaction of many effectors) or as a homeostatic regulation system. We explain that these formulations can be readily transformed into one another; particularly, the stable equilibrium (settling point) of the dynamical system is equivalent to the setpoint of the homeostatic formulation. Thus, we have tried to make it clear that setpoint vs. settling-point formulation is only a matter of the point of view.

Last but not least, one example where we could have discussed the evidence supporting the homeostatic role of the hypothalamus is in thermoregulation (see the text below). However, we did not see how to include it in the manuscript, without straying too far from issues relevant to the novel contributions of the paper.

A particularly prominent example for the role of the hypothalamus in homeostatic regulation has come over the years in the human and animal thermoregulation literature. Interestingly, the concept of an internally regulated set point appears prominently in that body of literature. The classical review by Benzinger (2) establishes experimental evidence for a thermal set point as a physiological property and points out the role of the “preoptic-spraoptic region of the hypothalamus” in central regulation of the body temperature. The suggestion that hypothalamic circuits play a role in maintenance of thermal homeostasis by translating sensed temperature into neural activity was formalized by Hammel (3) in a model proposing hypothalamic circuitry where integration of thermo-sensitive and thermo-insensitive neuronal activities could lead to dynamic encoding of the thermal setpoint. Populations of heat-sensitive neurons have been identified in the hypothalamus (4, 5): they increase firing rate with increasing body temperature. The heat-sensitive (HS) and heat-insensitive (HI) neurons synaptically innervate two sets of effector neurons. Heat-loss effectors are excited by the heat-sensitive cells and inhibited by the heat-insensitive cells in a manner that balances these inputs at 37C body temperature. Heat-production effector neurons are in turn inhibited by the HS neurons and excited by the HI neurons. Over the years, heat-loss/production effector neurons have been electrophysiologically identified (6–8) and anatomically mapped. The regulatory loop is closed by the thermosensory afferents from the periphery to the HS (but not the HI) neurons (e.g. see (9)). Manipulations of the POA induce body temperature changes (e.g. see (10)) and the effector neurons have been implicated in control of organismal thermoregulatory responses (e.g. as reviewed in Morrison and Nakamura (11) and Boulant (12)) including shivering (13). Furthermore, there is experimental evidence that the regulated thermal temperature point is influenced (or dynamically set) by signals that not in themselves directly related to temperature (e.g. hormonal levels, inputs from joint mechano-receptors receptors) and varies from individual to individual (see (12) for review).

*A distinction between tonic and phasic dopamine activity needs to be made*.

To make a distinction between tonic and phasic dopamine, we clearly stated in the new manuscript that our model, as in the classical RL model, only addresses the burst (i.e. phasic) activity pattern of dopamine neurons. How changes in the tonic DA levels might be incorporated into our theory is an active topic of current research in our group.

Reviewer 3:

*The theory is based on abstracted concepts like homeostatic set point and distances in the homeostatic space that do not directly correspond to any biological properties*.

Indeed, our mathematical theory, as any mathematical theory, requires several constructs to be defined. Above, in the response to reviewer 2, we discussed how the concepts used in our framework relate to ideas of dynamically-maintained internal equilibrium and potentials of dynamical systems. The functional equivalency of the two approaches establishes a correspondence between their neurobiological implementation. Thus, we respectfully beg to differ with the opinion of the reviewer, we believe that concepts we use do have connections to biological properties. For example the homeostatic space is simply a coordinate system where the various physiologically regulated quantities are represented: temperature, glucose levels, etc. Also, the setpoint is just equivalent to the stable equilibrium of the underlying dynamical system.

Let us take the example of temperature. Without going into details, as discussed in the text in response to the second reviewer, there are multiple classes of temperature receptors peripherally, and temperature sensitive neurons in the hypothalamus. There have been data-driven suggestions in the literature that activity of such neurons, informed by peripheral afferents, together with temperature insensitive neurons in the hypothalamus, encode the thermal setpoint (approx. 37 degrees) (3–5). There is further evidence that inputs from such neurons, create cold-producing and heat-producing effector neurons (6–8). Modern work on human thermoregulation experimentally suggests an existence of temperature space and “energy functional”, or in our terms, drive function (14).

*The model assumes that organisms experience all paths in the homeostatic space, and only then can choose the shortest path. However, once the animal reaches an extreme homeostatic deviation, it can never return (due to death)*.

Indeed we thank the reviewer for pointing out that we needed to clarify this point. We added a new subsection titled “Stepping back from the brink”, and addressed this issue in detail. In fact our model predicts that animals should learn to act preventively to avoid states with drastic deviations (even without experiencing them directly).

*The authors claim to provide a normative explanation for temporal discounting for the first time. However, alternative factors such as the environment being less and less predictable as one moves to the future seem to be a plausible explanation of temporal discounting*.

Indeed we agree with the reviewer that temporal discounting intuitive sense for a number of reasons including uncertainty of outcomes in the future, changing environments, etc. However the point we attempted to make was more formal and mathematical: if we were not to include discounting, behavioral policies that maximized rewards did not necessarily minimize the total deviation from the homeostasis and hence could endanger the animal. Hence lack of discounting did not result in equivalence of reward maximization of homeostatic defense. Temporal discounting ensured that such did not happen and ensured the rationality of defending homeostasis. In view of the reviewers comments, we realized that our claim was overreaching and withdrew the claim that our normative explanation for temporal discounting is the only possible explanation. Though, we have not been able to find any alternative formal mathematical explanation.

*Quoting form the review: I question whether the theory is falsifiable, as I do not see specific testable predictions*.

We thank the reviewer for pointing us toward this point. We added a subsection titled “predictions”, and listed five testable predictions of the model.

*Throughout the results I would have preferred either more incorporation of the equations or stronger references to the methods*.

We tried to incorporate more formal details in the text, particularly in the development of the theory. At the same time, we felt that including the full mathematical proofs in the main text of the paper would make it too cumbersome. These are now in the Materials and Methods section.

*When free parameters in the model are fit to one data set, they should show they can reproduce other data without extra fitting*.

We thank the reviewer for this comment. It should be mentioned that the different experimental data we have replicated in the paper come from different species (rat in the anticipatory responding task, and pigeon in the oral/fistula water-seeking task). Thus, it is not surprising that the free parameters have different values for different experimental data sets. For every individual dataset, however, the value of free parameters are chosen to replicate the first part of data, and then the same values have successfully predicted the second part. That is, for the case of anticipatory responding simulations, the free parameters are derived according to the training days (the first 8 days of the experiment), and then are used for predicting the extinction days, as well as the re-acquisition day. Similarly, for the case of oral/fistula water-seeking experiment, the free parameters are chosen to best explain the reinforcement experiment (Figure 7), and are then used for predicting the satiation experiment (Figure 8).

It is also noteworthy that although free parameters are different across different experiments (different species), the essential patterns of simulation results hold for a wide range of free parameters, and the specific values used in every experiment are only to replicate that specific data.

References:

1) McClure SM, Daw ND, Montague PR (2003) A computational substrate for incentive salience. Trends in Neurosciences 26:423–428.

2) Benzinger TH (1961) The diminution of thermoregulatory sweating during cold-reception at the skin. Proceedings of the National Academy of Sciences of the United States of America 47:1683–8.

3) Hammel H (1965) in Physiological Controls and Regulations, eds Yamamoto W, Brobeck J (Saunders, Philadelphia, PA), pp 71–97.

4) Makayama T, Elisenman JS, Hardy JD (1961) Single unit activity of anterior hypothalamus during local heating. Science 134:560–1.

5) Griffin JD, Kaple ML, Chow AR, Boulant JA (1996) Cellular mechanisms for neuronal thermosensitivity in the rat hypothalamus. The Journal of physiology 492 ( Pt 1:231–42.

6) Edinger HM, Eisenman JS (1970) Thermosensitive neurons in tuberal and posterior hypothalamus of cats. The American journal of physiology 219:1098–103.

7) Curras MC, Kelso SR, Boulant JA (1991) Intracellular analysis of inherent and synaptic activity in hypothalamic thermosensitive neurones in the rat. The Journal of physiology 440:257–71.

8) Dean JB, Boulant JA (1989) Effects of synaptic blockade on thermosensitive neurons in rat diencephalon in vitro. The American journal of physiology 257:R65–73.

9) Cliffer KD, Burstein R, Giesler GJ (1991) Distributions of spinothalamic, spinohypothalamic, and spinotelencephalic fibers revealed by anterograde transport of PHA-L in rats. The Journal of neuroscience :theofficial journ al of the Society for Neuroscience 11:852–68.

10) Chen XM, Hosono T, Yoda T, Fukuda Y, Kanosue K (1998) Efferent projection from the preoptic area for the control of non-shivering thermogenesis in rats. The Journal of physiology 512 ( Pt 3:883–92.

11) Morrison SF, Nakamura K (2011) Central neural pathways for thermoregulation. Frontiers in bioscience (Landmark edition) 16:74–104.

12) Boulant JA (2006) Neuronal basis of Hammel’s model for set-point thermoregulation. Journal of applied physiology (Bethesda, Md: 1985).100:1347–54.

13) Zhang YH, Yanase-Fujiwara M, Hosono T, Kanosue K (1995) Warm and cold signals from the preoptic area: which contribute more to the control of shivering in rats? The Journal of physiology 485 ( Pt 1:195–202.

14) Kingma BR, Frijns AJ, Schellen L, Van Marken Lichtenbelt WD (2014) Beyond the classic thermoneutral zone: Including thermal comfort. Temperature 1:142–149.

*[Editors' note: further revisions were requested prior to acceptance, as described below*.*]*

*Reviewer #1*:

*The authors have embarked on the valuable task of producing a computational framework that combines theories of reinforcement learning with those of homeostasis and drive reduction. This is a worthwhile goal and the authors have several examples of behaviors that arise within their framework as well as predictions. I do think the manuscript reads a bit as though come of the ideas of combining reinforcement learning and homeostasis are novel to the authors, whereas in reality their contribution is to add a mathematical/computational framework which allows for quantitative predictions to be made and suggests what could/should be observed in any neural mechanism*.

*While overall the writing is very clear, I think the manuscript would be served by the authors being more careful to tone down statements that suggest the idea of combining homeostasis and reinforcement learning is their own. After all, everyone knows that when one is out on a cold winter's day a hot drink is rewarding, whereas in the middle of a hot summer's day a cold drink has greater rewarding value. The authors deserve credit for developing a mathematical scheme (the first I think?) where such results fall out, and I think they now have enough quantitative results and predictions that make the scheme testable*.

In response to the reviewer’s suggestion, we added to the paragraph where we first talk about the contributions of the paper (in Introduction):

Given this evident coupling of homeostatic and learning processes, here, we propose a formal hypothesis for what computations, at an algorithmic level, may be performed in this biological integration of the two systems. More precisely, inspired by previous descriptive hypotheses on the interaction between motivation and learning (Hull, 1943; [35]; [57]), we suggest a principled model for how the rewarding value of outcomes is computed as a function of the animal’s internal state, and of the approximated need-reduction ability of the outcome…

Also, we added the below sentence to the conclusion section:

Being inspired by the classic drive-reduction theory of motivation, our mathematical treatment allows for quantitative results to be obtained, predictions that make the theory testable, and logical coherence.

*In a similar vein, some statements to motivate the work are exaggerated, for example in the first line of Discussion the authors’ state*:

*“Theories of conditioning are founded on the argument that animals seek reward, while reward is defined as what animals seek*.*”*

*I think that while these definitions can be found, to state simply “reward is defined” without adding “by some” or “can be defined” or “has been defined by some” is too bold and general*. *One can find plenty of definitions of reward, in which “primary reward” is “that which aids survival” or “helps propagate the species” or simply in general English, reward is something that is good for you!*

In response to the reviewer’s concern, we added the phrase “at least in the behaviorist approach” to the mentioned sentence:

Theories of conditioning are founded on the argument that animals seek reward, while reward is defined, at least in the behaviorist approach, as what animals seek.

*In a couple of places (including the Abstract) the authors state that they*:

*“prove analytically that reward-seeking and physiological stability are two sides of the same coin” and “Our theory mathematically proves that seeking rewards is equivalent to the fundamental objective of physiological stability” whereas in fact through their definition of drive;“we define the “drive” as the distance of the internal state from the setpoint” the authors assume this to be the case and develop a mathematical theory where this result is true. One must be careful in mathematical proofs as to what are the premises. Since the rewards associated with sexual desire are outside the model (as the authors comment) it is clear that it is only within their theory that the mathematical “proof” holds*.

In response to the reviewer’s concern, we added the phrase “Within this framework,” in the Abstract:

Within this framework, we mathematically prove that seeking rewards is equivalent to the fundamental objective of physiological stability, defining the notion of physiological rationality of behavior.

Furthermore we added the phrase “On the basis of the proposed computational integration of the two systems” into the sentence below, in the Introduction section:

On the basis of the proposed computational integration of the two systems, we prove analytically that reward-seeking and physiological stability are two sides of the same coin, and also provide a normative explanation for temporal discounting of reward.

*Reviewer #2*:

*1) The authors make a comment early on that equates reward/reinforcer/utility. Given the obvious sophistication of the authors, this is unfortunate. In particular, to make clear the relationship between prior treatments of utility and the authors’ proposal would be helpful. Notably, the authors do describe other approaches to this, but even a sentence or two early on that clarifies rather than lumps together the difference between reinforcer/utility. Specifically because the authors are essentially arguing that homeostatic utility determines reinforcement properties*.

We thank the author for pointing out this issue. By “utility”, we mean “economic utility” (as it is defined in Economics) rather than “homeostatic utility”. In economics, the utility of a commodity is a fixed value, without taking the internal state of individuals into account. This is the same problem as with reinforcer/reward value in psychology. In order to resolve this misunderstanding, we now use the term “economic utility” rather than “utility”, in the manuscript.

*2) The authors make a comment about 'erroneous estimation of error' and later in the manuscript talk at length about, essentially, taste serving as cues. Three lines of investigation that the authors might find useful in this discussion: (1) Beeler et al Eur J Neuroscience 2012 'taste uncoupled from nutrition fails to sustain the rewarding properties of . . . ' (2) the work of Swithers with artificial sweeteners*:

*Swithers, S.E. & Davidson, T.L. (2008) A role for sweet taste: calorie predictive relations in energy regulation by rats. Behav. Neurosci., 122, 161- 173*.

*Swithers, S.E., Baker, C.R. & Davidson, T.L. (2009) General and persistent effects of high-intensity sweeteners on body weight gain and caloric compensation in rats. Behav. Neurosci., 123, 772-780*.

*Swithers, S.E., Martin, A.A. & Davidson, T.L. (2010) High-intensity sweeteners and energy balance. Physiol. Behav., 100, 55-62*.

*Finally, the authors cite one paper by de Araujo, but he has significantly developed the notion that the DA cells specifically serve as a metabolic sensor*.

We found these references very helpful in supporting some aspects of our theory. In this respect, we added the below paragraph to the end of the subsection “Neural substrates”:

Such orosensory-based approximation of nutritional content, could have been obtained through evolutionary processes (6), as well as through prior learning (2; 60, 61). In the latter case, approximations based on orosensory or contextual cues can be updated so as to match the true nutritional value, resulting in a rational neural/behavioral response to food stimuli (De Araujo et al., 2008).

The last sentence suggests a probable mechanism for the taste-independent adaptation of dopamine response to the true caloric value of food.

*Other than that, I think there are many things that one could nitpick about, especially with regards to the endless details and nuances of the model (eg., I am not sure the authors have fully addressed the question the other reviewer had regarding the 'shortest distance between two points' idea). However, I think the paper is interesting, brings up some very good points, is well done and, as the authors point out, targets the mutual weakness of HR and RL models and brings them together nicely*.

*Reviewer 3*:

*1) Scientifically, I think you need to highlight and unpack the major result in the appendix. At an appropriate point in the main text, I would include a paragraph of the following sort*:

*“In summary, we have established a formal link between the homeostatic imperatives to keep physiological states near some set point and the maximisation of temporally discounted reward (or minimisation of some loss function). This is an important and non-trivial result. The appendix provides a formal proof; however, the underlying idea is fairly simple. Imagine you had to plan a hill walk, during which you wanted to maximise the height (altitude or reward) averaged over the path you take. If someone dropped you at the bottom of the hill, the optimum path would be to ascend the hill and spend as long as possible at the top before returning to your pick up point. Notice that this entails ascending the hill (reward function) before descending. Implicit in this strategy is a maximisation of temporally discounted reward. In other words, going up the hill first and then coming down is better than going down and then coming back up. It is this fundamental (variational) phenomenon that connects homeostasis with classical temporal discounting*.

*Furthermore, as indicated above, if the homeostatic cost (negative reward) is cast as a log probability then it can be treated as (free) energy*.

Thanks to the reviewer’s suggestion, we now explain the importance of temporal discounting more clearly by adding the paragraph below (modified version of the paragraph suggested by the reviewer) in the middle of the section “Normative role of temporal discounting”:

Imagine you had to plan a 1-hr hill walk from a drop-point toward a pickup point, during which you wanted to minimize the height (equivalent to drive) summed over the path you take. In this summation, if you give higher weights to your height in the near future as compared to later times, the optimum path would be to descend the hill and spend as long as possible at the bottom (i.e. homeostatic setpoint) before returning to the pickup point. [Disp-formula equ5] shows that this optimization is equivalent to optimizing the total discounted rewards along the path, given that descending and ascending steps are defined as being rewarding and punishing, respectively ([Disp-formula equ2]).

In contrast, if at all points in time you give equal weights to your height, then the summed height over path only depends on the drop and pickup points, since every ascend can be compensated with a descend at any time.

We chose not to include the second part of the suggested paragraph: with all due gratitude for the reviewers support of our work and appreciation for the efforts of the reviewer to help us improve the clarity of the paper, we felt that launching into a short discussion of the free-energy principle early in our manuscript, before we sowed out the major results of the paper, would be distracting to the reader. We give ample discussion of the relationship between our theory and the free-energy principle in the Discussion where we point out exactly what the reviewer urges us to highlight.

*Crucially, the time average or path integral of energy is called action. This means that both the homeostasis and temporally discounted reward are ways of prescribing a principle of least action. From this perspective, one can regard the adaptive behaviours that we are trying to link as necessary and emergent properties of all dynamical systems that comply with (Hamilton's) principle of least action. We will return to this perspective in the Discussion*.*”*

We thank the reviewer for the suggested texts to be added to the manuscript. We used some of the notions mentioned by the reviewer (particularly the principle of least action), and discussed them in the manuscript. For example we added the below text after [Disp-formula equ14]:

The equivalency of reward maximization and physiological stability objectives in our model ([Disp-formula equ5]) shows that optimizing either homeostasis or sum of discounted rewards corresponds to prescribing a principle of least action applied to the surprise function.

*2) The second major point is about the format of your paper. It is still unclear where the reader can find the details of your simulations. I also note that you have included supplementary figures. Can I suggest that you remove all supplementary material and place it in the main text (or discard it and refer to it as results not shown). I think you should prepare the reader for the slightly unusual scientific presentation with a paragraph at the beginning of the paper along the following lines*:

*“We will develop our theoretical results by appealing to simulations. These simulations are described in figures (and accompanying tables) and are called upon when necessary. All the simulations in this paper followed the same procedure: first we define a model that captures the problem of interest in terms of a Markov decision process. The ensuing behaviour is then optimised using classical reinforcement learning procedures (Q-learning) to define a value function. Actions are then selected using a softmax function of the value of allowable actions or choices. For each simulation we present the graphical model or Markov decision process in the figures, along with the ensuing behaviour. Each figure is accompanied by a table specifying the parameters of the Markovian process, the Q-learning and softmax functions used to simulate behaviour*.*”*

*Note that I am suggesting, for every simulation you present, a figure and table. Whenever you refer to results that are not presented in this format I would say so explicitly so the reader does not have to wonder whether they have missed something*.

In order to give a better outline of the structure of the paper, we changed the last paragraph of the Introduction section to this:

The paper is structured as follows: After giving a heuristic sketch of the theory, we show several analytical, behavioral, and neurobiological results. On the basis of the proposed computational integration of the two systems, we prove analytically that reward-seeking and physiological stability are two sides of the same coin, and also provide a normative explanation for temporal discounting of reward. Behaviorally, the theory gives a plausible unified account for anticipatory responding and the rise-fall pattern of the response rate. We show that the interaction between the two systems is critical in these behavioral phenomena and thus, neither classical RL nor classical HR theories can account for them. Neurobiologically, we show that our model can shed light on recent findings on the interaction between the hypothalamus and the reward-learning circuitry, namely, the modulation of dopaminergic activity by hypothalamic signals.

Furthermore, we show how orosensory information can be integrated with internal signals in a principled way, resulting in accounting for experimental results on consummatory behaviors, as well as the pathological condition of over-eating induced by hyperpalatability.

Finally, we discuss limitations of the theory, compare it with other theoretical accounts of motivation and internal state regulation, and outline testable predictions and future directions.

Furthermore, we moved “Figure 4–figure supplements 2, 3 and 4” in the previous manuscript into the main text in the current version of the manuscript (merged together in Figure 4).

Also, in order to provide more details of the simulations and to have the same format for all presented results (i.e., problem definition, simulation results, simulated environment (MDP), free parameters of the model), we added four tables (Figure 5–figure supplement 1; Figure 6—figure supplement 2; Figure 10–figure supplement 1; Figure 12—figure supplement 1) and one Markov Decision Process (Figure 12–figure supplement 2) in the figure supplements.